# Ice nucleation activity of silicates and aluminosilicates in pure water and aqueous solutions. Part I - The K-feldspar Microcline

Anand Kumar, Claudia Marcolli, Beiping Luo, Thomas Peter

Institute for Atmospheric and Climate Sciences, ETH Zurich, Zurich, 8092, Switzerland
5    *Correspondence to:* Anand Kumar (anand.kumar@env.ethz.ch)

**Abstract.** Potassium containing feldspars (K-feldspars) have been considered key mineral dusts for ice nucleation (IN) in mixed-phase clouds. To investigate the effect of solutes on their IN efficiency, we performed immersion freezing experiments with the K-feldspar microcline, which is highly IN active. Freezing of emulsified droplets with microcline suspended in aqueous solutions of $NH_3$, $(NH_4)_2SO_4$, $NH_4HSO_4$, $NH_4NO_3$, $NH_4Cl$, $Na_2SO_4$, $H_2SO_4$, $K_2SO_4$ and $KCl$, with solute concentrations corresponding to water activities $a_w = 0.9 - 1.0$, were investigated by means of a differential scanning calorimeter (DSC). The measured heterogeneous IN onset temperatures, $T_{\text{het}}(a_w)$ deviate strongly from $T_{\text{het}}^{\Delta a_w^{\text{het}}}(a_w)$, the values calculated from the water-activity-based approach (where $T_{\text{het}}^{\Delta a_w^{\text{het}}}(a_w) = T_{\text{melt}}(a_w + \Delta a_w^{\text{het}})$ with a constant offset $\Delta a_w^{\text{het}}$ with respect to the ice melting point curve). Surprisingly, for very dilute solutions of $NH_3$ and $NH_4^+$-salts (molalities $\lesssim 1$ mol kg$^{-1}$ corresponding to $a_w \gtrsim 0.96$), we find IN temperatures raised by up to 4.5 K above the onset freezing temperature of microcline in pure water ($T_{\text{het}}(a_w = 1)$) and 5.5 K above $T_{\text{het}}^{\Delta a_w^{\text{het}}}(a_w)$, revealing $NH_3$ and $NH_4^+$ to significantly enhance the IN of the microcline surface. Conversely, more concentrated $NH_3$ and $NH_4^+$ solutions show a depression of the onset temperature below $T_{\text{het}}^{\Delta a_w^{\text{het}}}(a_w)$ by as much as 13.5 K caused by a decline in IN ability accompanied with a reduction in the volume fraction of water frozen heterogeneously. All salt solutions not containing $NH_4^+$ as cation exhibit nucleation temperatures $T_{\text{het}}(a_w) < T_{\text{het}}^{\Delta a_w^{\text{het}}}(a_w)$ even at very small solute concentrations. In all these cases, the heterogeneous freezing peak displays a decrease as solute concentration increases. This deviation from $\Delta a_w^{\text{het}} = $ const. indicates specific chemical interactions between particular solutes and the microcline surface not captured by the water-activity-based approach. One such interaction is the exchange of $K^+$ available on the microcline surface with externally added cations (e.g. $NH_4^+$). However, the presence of a similar increase in IN efficiency in dilute ammonia solutions indicates that the cation exchange cannot explain the increase in IN temperatures. Instead, we hypothesize that $NH_3$ molecules hydrogen bonded on the microcline surface form an ice-like overlayer, which provides hydrogen bonding favorable for ice to nucleate on, thus enhancing both the freezing temperatures and the heterogeneously frozen fraction in dilute $NH_3$ and $NH_4^+$ solutions. Moreover, we show that aging of microcline in concentrated solutions over several days does not impair IN efficiency permanently in case of near neutral solutions since most of it recovers when aged particles are re-suspended in pure water. In contrast, exposure to severe acidity (pH $\lesssim 1.2$) or alkalinity (pH $\gtrsim 11.7$) damages the microcline surface, hampering or even destroying the IN efficiency irreversibly. Implications for IN on airborne dust containing microcline might be multifold, ranging from a reduction of immersion freezing when exposed to dry, cold and acidic conditions, to a 5-K enhancement during condensation freezing when microcline particles experience high humidity ($a_w \gtrsim 0.96$) at warm (252 - 257 K) and $NH_3/NH_4^+$-rich conditions.

## 1 Introduction

Ice clouds play an important role in the Earth's radiative budget and hence climate (IPCC, 2013). The radiative properties of an ice cloud is determined mainly by its optical depth and its temperature, i.e. altitude (Corti and Peter, 2009), which in turn depend on the number density and size of the ice crystals formed (Stephens et al., 1990; 2003a; DeMott et al., 2003b). Additionally, the formation of ice crystals affects cloud dynamics and in-cloud chemical processes, and it is one of the most effective pathways to form precipitation in mid-latitudes, thus affecting cloud lifetime (Lohmann, 2006). Given the existing uncertainties in understanding the formation of cold, high cirrus and warmer, lower mixed-phase clouds, it remains imperative to continue exploring the various ice nucleation (IN) mechanisms.

Ice formation in the atmosphere happens via both homogeneous and heterogeneous nucleation. Homogeneous IN may occur spontaneously as a stochastic process in an aqueous liquid droplet, which is metastable with respect to ice. For pure water droplets, homogeneous IN has been described by classical nucleation theory (see Ickes et al. (2015) for a recent review). For solution droplets, a thermodynamic approach by Koop et al. (2000) shows that the nucleation is independent of the nature of the solute, but depends only on the water activity ($a_w$) of the solution, such that the freezing point curve (for droplets of a certain size) can be derived from the melting point curve by a constant shift in $a_w$, i.e. $T_{\mathrm{het}}^{\Delta a_w^{\mathrm{hom}}}(a_w) = T_{\mathrm{melt}}(a_w + \Delta a_w^{\mathrm{hom}})$, where $\Delta a_w^{\mathrm{hom}} = 0.313$ (Koop and Zobrist, 2009). Statistical analysis of freezing experiments indicates that heterogeneous IN is induced by active sites present on foreign bodies termed ice-nucleating particles (INPs). Active sites are preferred locations for IN with minimum areas in the order of $10 - 50$ nm$^2$ based on estimates using classical nucleation theory (Vali, 2014; Vali et al., 2015; Kaufmann et al., 2017). IN can proceed through different mechanisms. Here we adopt the IN and freezing terminology summarized by Vali et al. (2015). *Immersion freezing* occurs when the INP is immersed in a supercooled water droplet. *Condensation freezing* is considered to take place when IN is concurrent with cloud droplet activation. Although condensation freezing is usually mentioned as a distinct mode, it is still debated whether, on a microscopic scale, it is really distinct from the other freezing modes (Vali et al., 2015). Furthermore, freezing can be initiated by an insoluble particle which penetrates the surface of a supercooled liquid droplet from the outside, called *contact freezing* (Pruppacher and Klett, 1994; Vali et al., 2015). This pathway could also occur if the particle penetrates the droplet surface from the inside out (Durant and Shaw, 2005; Shaw et al., 2005) and its efficiency depends on the position of the particle with respect to the droplet surface (Nagare et al., 2016). Finally, there is one mechanism considered to nucleate the ice phase directly by deposition of water vapor without forming liquid water as an intermediate, namely *deposition nucleation*. This mechanism was recently questioned by Marcolli (2014), who suggested that what is considered deposition nucleation is in most cases pore condensation and freezing (PCF) occurring in cavities on INPs which may retain water owing to surface curvature forces described by the inverse Kelvin effect.

Mineral dust particles, composed mainly of quartz, feldspars, calcite and clays viz. kaolinite, illite and montmorillonite in varying proportions, are a well-established class of INPs (Pruppacher and Klett, 1994; Zobrist et al., 2008; Lüönd et al., 2010; Murray et al., 2011; Hoose and Möhler, 2012; Pinti et al., 2012; Atkinson et al., 2013; Cziczo et al., 2013; Kaufmann et al., 2016). A study by Atkinson et al. (2013) performed on minerals from the clay group and also on K-feldspar, Na/Ca-feldspar, quartz, and calcite showed that the IN efficiency in the immersion mode of K-feldspar is exceptionally high compared to the other minerals. Droplets containing K-feldspar reached a frozen fraction of 50 % at 250.5 K compared with values below 240 K for the investigated clay mineral. However, not all K-feldspar polymorphs exhibit the same high IN efficiency. Microcline has been reported to be IN active at higher temperatures than orthoclase and sanidine (Kaufmann et al., 2016).

A large proportion of earth's crust consists of feldspar minerals, but only a minor fraction (~13% according to Murray et al. (2012)) of the airborne atmospheric dust aerosol is composed of feldspar mineral dust. However, considering that particles exceeding 1 µm in diameter are typically aggregates of different minerals (Reid and Sayer, 2003; Kandler et al., 2011), even a minor component of microcline might suffice to provide such supermicron particles with excellent active sites and hence a high IN efficiency (Kaufmann et al., 2016).

To assess the relevance of atmospheric dusts for cloud glaciation, the role of atmospheric processing for their IN efficiency needs to be taken into account. Nitrates, sulfates and organic compounds present in the atmosphere are prone to adhere to the mineral dust particle surface (Murphy and Thomson, 1997; Grassian, 2002). This process can take place already at the source or during long-range transport of aerosol particles. Such chemical coatings may change the particle surface and increase the hygroscopicity of the mineral dust core, e.g. making it easier for the particles to take up water and form cloud droplets (Levin et al., 1996; Herich et al., 2009). Since dust particles are an ensemble of various minerals with potentially different IN efficiencies, the aging process and physico-chemical surface modifications add another complication in resolving their IN efficiency (Prospero, 1999; DeMott et al., 2003b; Sassen et al., 2003). It is imperative to single out components of mineral dusts and study how general or specific coating/solute effects are in order to infer the role of atmospheric aging on IN by mineral dust particles.

Describing the kinetic (non-equilibrium) IN process as a simple function of a thermodynamic (equilibrium) quantity $\Delta a_w$ is tempting because of its simplicity. Therefore, the applicability of the "*water-activity criterion*" proposed by Koop et al. (2000) for homogeneous IN to heterogeneous IN temperatures of various types of INPs in immersion mode has been probed by several studies in the past. Zuberi et al. (2002) studied kaolinite and montmorillonite immersed in ammonium sulfate solution droplets using optical microscopy. They tested, whether the heterogeneous freezing on these clay minerals could be described by a constant water activity offset, i.e. $T_{het}^{\Delta a_{w,het}}(a_w) = T_{melt}(a_w + \Delta a_{w,het})$, and found general agreement for $\Delta a_w^{kaol,mont} = 0.242$, but with remaining discrepancies either due to experimental conditions or due to heterogeneous freezing not satisfying $\Delta a_w^{het} = \text{const}$. At this point it should be noted that the homogeneous IN approach by Koop et al. (2000) has a broad empirical basis and theoretical underpinning, whereas heterogeneous IN satisfying $\Delta a_w^{het} = \text{const}$ is only valid under the assumption that the solute does not affect the foreign surface.

Zobrist et al. (2008) used differential scanning calorimetry (DSC) and optical microscopy to observe immersion freezing of silver iodide, nonadecanol, silica and Arizona Test Dust (ATD) in solution droplets containing several organic substances and inorganic salts including ammonium sulfate and sulfuric acid. They found that, when the results are analyzed in terms of the solution $a_w$, a very consistent behavior emerged, allowing them to derive values for $\Delta a_w^{het}$, which described their measurements mostly within experimental error. Recently, Knopf and Alpert (2013) used an even wider variety of INPs and solution mixtures (Pahokee Peat and Leonardite serving as surrogates of Humic-Like Substances (HULIS); aqueous NaCl droplets containing two distinct species of phytoplankton *Nannochloris atomus* and *Thalassiosira pseudonana*; aqueous NaCl droplets coated by 1-nonadecanol; aqueous $(NH_4)_2SO_4$ droplets containing illite; kaolinite, aluminum and iron oxide particles, and fungal spores suspended in various aqueous solutions), leading them to conclude that $\Delta a_w^{het} = \text{const}$ provides a good description (with a different offset for each INP) and termed this the "$a_w$-based immersion freezing model (ABIFM)". Several other studies that have applied this approach to describe heterogeneous IN include Zobrist et al. (2006), Knopf et al. (2011) and Archuleta et al. (2005) (see Knopf et al. (2018) for a detailed review on this topic).

In contrast to $\Delta a_w^{het}$ = const, several other studies conducted with clay minerals and ATD coated with sulfuric acid (Cziczo et al., 2009; Chernoff and Bertram, 2010; Sullivan et al., 2010a; Augustin-Bauditz et al., 2014) reported a deterioration/inhibition of IN efficiency of dust particles caused by an organic and/or inorganic coating/treatment. The reduction in the IN efficiency could possibly be attributable to surface deterioration from acid treatment resulting in destruction of the active sites. Möhler et al. (2008) observed marked but gradual decrease in IN efficiency of ATD particles coated with secondary organic aerosol. In this case, active sites might remain intact but disabled to act as INP because of the organic cover. In the absence of surface chemical reactions, the chemical coating might dissolve during cloud droplet activation, thereby releasing the mineral surface again and restoring the ice-nucleating ability (Sullivan et al., 2010b; Tobo et al., 2012; Kulkarni et al., 2014; Wex et al., 2014).

Evidently, it is difficult to understand the interactions of solute coatings with an INP surface consisting of several minerals. In this and the companion papers (Part II and III) we attempt to relate IN activities of pure mineral surfaces with the mineral surface properties by investigating the differences in IN activity of structurally similar minerals in pure water and aqueous solutions. In this study we present immersion freezing experiments using a DSC on particles consisting of one single mineral, namely the most IN-active K-feldspar polymorph, microcline, dispersed in solution droplets containing sulfuric acid, ammonia and several inorganic salts viz. ammonium sulfate, ammonium chloride, ammonium nitrate, ammonium bisulfate, sodium sulfate, potassium chloride, potassium sulfate. We will see that microcline deviates markedly from $\Delta a_w^{het}$ = const.

## 2 Methodology

### 2.1 Mineralogy, size distribution and BET surface area measurements

Feldspars are crystalline aluminosilicates with the general formula $XAl_{1-2}Si_{3-2}O_8$, often written as $XT_4O_8$, where T is an atom in tetrahedral coordination with oxygen, i.e. Al or Si. X represents an alkali or alkaline earth metal, acting as a charge compensating cation. The feldspar crystal lattice is composed of corner sharing $AlO_4^-$ and $SiO_4$ tetrahedra linked in an infinite 3D framework. Each of the four oxygen atoms in one tetrahedron is shared by the neighboring tetrahedra. Hence, the tetrahedron with Al at the center carries a single negative charge, which is compensated by $K^+/Na^+$ (for 1 Al atom) or $Ca^{2+}$ (for 2 Al atoms). Microcline is a K-rich feldspar and belongs together with sanidine and orthoclase to the potassium feldspar group.

We used a microcline sample from Macedonia provided by the Institute of Geochemistry and Petrology of ETH Zurich and milled it with a tungsten carbide disc mill. We determined the mineralogical composition of the milled feldspar sample by means of X-ray diffraction (XRD) in order to assess the mineralogical purity of the mineral. A quantitative analysis was performed with the AutoQuan program, a commercial product of GE Inspection Technologies applying a Rietveld refinement (Rietveld, 1967, 1969). Based on the X-ray diffractogram, the microcline sample consists of 86.33% (± 1.71%) microcline, mixed with orthoclase (6.18% ± 0.72%) and albite (7.49% ± 0.48%), a Na-rich feldspar. Number size distribution of the milled mineral was obtained with a TSI 3080 scanning mobility particle sizer (SMPS) and a TSI 3321 aerodynamic particle sizer (APS), and merged as described by Beddows et al. (2010). A lognormal distribution was fitted to the size distribution yielding a mode diameter of 213 nm (Supplementary Material, Fig. S17). The Brunauer–Emmett–Teller (BET) nitrogen adsorption method was used to determine the specific surface area for microcline as 1.91 $m^2\,g^{-1}$.

### 2.2 Immersion freezing with emulsions of freshly prepared microcline suspensions in water or aqueous solutions

Immersion freezing experiments were carried out with a DSC (Q10 from TA instruments) (see Zobrist et al. (2008) for details). We prepared microcline suspensions of 2 wt% in water (molecular biology reagent water from Sigma-Aldrich) with varying solute

concentrations (0 – 15 wt%) viz. $H_2SO_4$ (Sigma Aldrich, 96.5%), $NH_3$ solution (Merck, 25%), $(NH_4)_2SO_4$ (Sigma Aldrich, ≥ 99%),

NH$_4$Cl (Sigma Aldrich, ≥ 99.5%), $NH_4NO_3$ (Fluka, ≥ 99.5%), $NH_4HSO_4$ (Fluka, ≥ 99.5%), $Na_2SO_4$ (Sigma Aldrich, ≥ 99%), KCl (Sigma Aldrich, ≥ 99%), $K_2SO_4$ (Sigma Aldrich, ≥ 99%). To avoid particle aggregation, we sonicated microcline suspensions prepared in pure water or salt solutions for 5 min before preparing the emulsions. The aqueous suspension and an oil/surfactant mixture (95 wt% mineral oil (Sigma Aldrich) and 5 wt% lanolin (Fluka Chemical)) were mixed in a ratio of 1:4 and emulsified with a rotor-stator homogenizer (Polytron PT 1300D with a PT-DA 1307/2EC dispersing aggregate) for 40 seconds at 7000 rpm. This procedure leads to droplet size distributions peaking at about 2 – 3 µm in number and a broad distribution in volume with highest values between 4 and 12 µm similar as the ones shown in Figs. 1 of Marcolli et al. (2007), Pinti et al. (2012), and Kaufmann et al. (2016). Next, 4-10 mg of this emulsion was placed in an aluminum pan, hermetically closed, and then subjected to three freezing cycles in the DSC following the method developed and described by Marcolli et al. (2007). The first and the third freezing cycles were executed at a cooling rate of 10 K min$^{-1}$ to control the stability of the emulsion. We ran the second freezing cycle at 1 K min$^{-1}$ cooling rate and used it for evaluation of freezing temperature ($T_{het}$ and $T_{hom}$) and heterogeneously frozen fraction ($F_{het}$) as discussed below (Zobrist et al., 2008; Pinti et al., 2012; Kaufmann et al., 2016). Emulsions were always freshly prepared before a DSC experiment, which took about 75 - 90 min for the three freezing cycles depending on the investigated temperature range.

Figure 1 shows a typical DSC thermogram. The first peak occurring at a higher temperature shows the heat release due to heterogeneous freezing and the second peak occurring at a lower temperature is due to homogeneous freezing. Throughout this article, we define the freezing temperatures ($T_{het}$ and $T_{hom}$) as the onset points of the freezing signals (i.e., intersection of the tangents at the greatest slope of the freezing signal and the extrapolated baseline), whereas the melting temperature ($T_{melt}$) was determined as the minimum of the ice melting peak. The heat release is approximately proportional to the volume of water that froze heterogeneously or homogeneously and is represented by the integral of the peak over time. Since the enthalpy of freezing is temperature-dependent, this proportionality (as stated above) is only approximate (Speedy, 1987; Johari et al., 1994). We define the "heterogeneously frozen fraction", $F_{het}$, as the ratio of the heterogeneous freezing signal to the total freezing signal from water frozen heterogeneously and homogeneously. Spikes that may occur before the appearance of the heterogeneous freezing signal are excluded from the evaluation as they originate from particularly large single droplets in the tail of the droplet size distribution, which are not representative for the sample. We prepared at least two separate emulsions from each suspension and report mean values. Average precisions in $T_{het}$ are ±0.15 K with maximum deviations not exceeding 0.8 K. $T_{hom}$, and $T_{melt}$ are precise within ±0.1 K. Absolute uncertainties in $F_{het}$, are on average ± 0.02 and do not exceed ± 0.1.

**2.3 Immersion freezing experiments with microcline emulsions as a function of aging**

Microcline (2 wt%) suspended in pure water, ammonia solution (0.05 molal), and ammonium sulfate solutions (0.1 wt% and 10 wt%) were aged over a period of one week. Immersion freezing experiments as described in Section 2.2 were carried out with the DSC setup on the day of preparation (fresh), then on the first, fourth and seventh day after preparation in order to assess the long-term effect of ammonia and ammonium containing solutes on the IN efficiency of microcline. Each suspension was sonicated for 5 min before preparing the emulsions in order to re-suspend the particles that have settled over time and avoid particle aggregation.

**2.4 Reversibility of interactions between microcline and solutes tested in immersion freezing experiments**

Suspensions of 2 wt% microcline prepared with relatively high solute concentration — 10 wt % $(NH_4)_2SO_4$, 2 wt % $NH_4HSO_4$, 0.5 wt % $K_2SO_4$, 2 molal ammonia solution — were aged for 10 days. The aged suspensions were then centrifuged for 2 minutes at 600 rpm, the supernatant solution was removed and the settled particles were washed with pure water. This process was repeated

five times and the washed particles were resuspended either in pure water or in a dilute solution of the same solute (i.e. water, 0.1 wt % $(NH_4)_2SO_4$, 0.5 wt % $NH_4HSO_4$, 0.05 wt % $K_2SO_4$, 0.05 molal $NH_3$ solution). Using DSC, we compared immersion freezing of emulsions containing dust treated in this manner with emulsions of fresh dust prepared with the same solute concentration or in pure water.

## 3 Results

### 3.1 Effect of microcline concentration on the heterogeneous freezing signal

Figure 2 shows the DSC thermograms of the slow cooling run (1 K min$^{-1}$) performed on emulsions containing varying concentrations of microcline in pure water. The onset of the heterogeneous freezing signal at 251 K for the lowest microcline concentration shifts only slightly to 252 K for the highest concentration. The consistency of the freezing temperature indicates that microcline contains a prevalent nucleation site with a well-constrained IN temperature. The homogeneous freezing signal results either from the freezing of empty water/solution droplets or droplets containing particles which are IN inactive. With lower particle concentration, more droplets are devoid of particles hence contributing more to the homogeneous freezing signal. Higher concentrations lead to an increase in number of particles per droplet. The dependence of the heat signals on dust concentration was analyzed in more detail by Kaufmann et al. (2016). The median droplet diameter in the emulsion is ~2 µm. For 20 wt% microcline suspensions, droplets with diameters of 1.1 µm are on average filled with one microcline particle, while smaller droplets are on average empty. This number shifts to 6.3 µm for 0.2 wt% microcline suspensions. 2 wt% suspensions were picked for further investigations leading to a good heterogeneous freezing signal with little particle agglomeration (see also Kaufmann et al. (2016) and Appendix B). Droplets with diameters of about 12 µm are considered to be relevant for the freezing onset in the DSC. For a 2 wt% suspension of microcline, these droplets contain about 1000 microcline particles.

### 3.2 Dependence of the heterogeneous freezing temperatures and volume fractions on water activity

Figure 3 shows heterogeneous ($T_{het}$) and homogeneous freezing onsets ($T_{hom}$) and ice melting temperatures ($T_{melt}$) of all investigated solutes as a function of the solution water activity ($a_w$), which equals the ratio of vapor pressure of water in the solution and saturation vapor pressure of pure water under the same conditions. The $a_w$ is obtained from the melting point depression measured during the heating cycle using the Koop et al. (2000) parameterization. Hence all melting temperatures lie exactly on the melting curve, except in case of $Na_2SO_4$ where $a_w$ has been calculated based on the solute concentration using the AIOMFAC thermodynamic model at 298 K (Zuend et al., 2008; Zuend et al., 2011). This different procedure was necessary because above the eutectic concentration of $Na_2SO_4$ (> 4.6 wt%), a hydrate of $Na_2SO_4$ crystallizes together with ice (Negi and Anand, 1985) thus corrupting the $a_w$ determination based on the experimental melting temperatures. Although the water activity of solutions may show a temperature dependence, the values determined at the melting temperature were assumed valid also at the freezing temperatures. Such temperature dependencies vary from solute to solute (Ganbavale et al., 2014) and may explain the deviation of the measured homogeneous freezing points from the homogeneous freezing curve (dotted black line in Fig. 3) obtained by a constant shift of the melting curve (Koop et al., 2000) by $\Delta a_w^{hom}$ (T) = 0.296. This mean offset value is based on the present dataset and was obtained by a least-square root averaging of the individual $\Delta a_w^{hom}$ values of all measurements calculated as

$$\Delta a_w^{hom}(T) = a_w^{hom}(T) - a_w^{melt}(T) \tag{1}$$

where $a_w^{melt}(T)$ is the ice melting curve (Koop et al., 2000).

Similarly, a constant offset $\Delta a_w^{het} = 0.187$ is applied to shift the ice melting curve to the heterogeneous freezing temperature of pure water, yielding the solid black line, which will be referred to as $T_{het}^{\Delta a_w^{het}}(a_w)$ from here onwards for simplicity. This curve would be expected in the absence of specific interactions between the solute and the ice-nucleating surface so that the only effect of the solute is a freezing point depression. However, as can be seen from Fig. 3 the measured heterogeneous freezing onset temperatures, $T_{het}$, deviate from $T_{het}^{\Delta a_w^{het}}(a_w)$. For all $NH_4^+$ solute cases in dilute concentrations viz. $NH_3$ (< 1 molal), $(NH_4)_2SO_4$ (< 0.16 molal), $NH_4HSO_4$ (< 0.18 molal), $NH_4NO_3$ (< 0.67 molal) and $NH_4Cl$ (< 0.99 molal), there is an increase in $T_{het}$ compared to $T_{het}^{\Delta a_w^{het}}(a_w)$ near $a_w \sim 1$. On the other hand, even low concentrations of a non-$NH_4^+$ solute leads to strong decrease in $T_{het}$ compared to $T_{het}^{\Delta a_w^{het}}(a_w)$. The decrease in $T_{het}$ observed at lower $a_w$ varies depending on the solute. Figure 3 also depicts representative variations (maximum and minimum values) in $T_{het}$ and $a_w$ by vertical and horizontal bars, observed over several separate emulsion freezing experiments for each solute strength.

Variation in the intensities of heterogeneous and homogeneous freezing signals were also observed depending on solute concentration. Figure 4 shows the DSC thermograms for emulsion freezing of microcline suspended in $NH_4HSO_4$. It depicts that $T_{het}$ first increases then decreases while the area under the heterogeneous freezing signal continually decreases as the $NH_4HSO_4$ concentration increases. Figure 5 shows the variation of the heterogeneously frozen fraction $F_{het}$ (ratio of heterogeneous freezing signal to total freezing signal) with respect to $a_w$. For all $NH_4^+$ solute cases, $F_{het}$ shows an increase in very dilute solutions, with $NH_4HSO_4$ being the only exception. Even low concentrations of a non-$NH_4^+$ solute decrease $F_{het}$. The decrease in $F_{het}$ observed at higher $a_w$ varies depending on the solute. These trends are very similar to the ones observed for $T_{het}$ with a few exceptions. For the $NH_4^+$ solute cases, the $a_w$ range with an increase in $F_{het}$ is narrower than the range with an increase in $T_{het}$. The ammonia solutions show a higher enhancement of $F_{het}$ over a larger $a_w$ range compared to the $NH_4^+$-containing solutes.

## 4 Discussion

### 4.1 Heterogeneous ice nucleation of microcline in pure water

Our microcline sample (from Macedonia) exhibits onset freezing temperatures $T_{het}$ of 251 – 252 K for suspensions in pure water with concentrations between 0.2 wt% and 20 wt%. Kaufmann et al. (2016) investigated two microcline samples with onset freezing temperatures of 250.5 – 251.8 K (microcline from Namibia) and 251.8 – 252.8 K (microcline from Elba) for suspension concentrations of 0.5 – 10 wt% . Moreover, they determined active particle fractions of 0.64 for Microcline Elba and 0.54 for Microcline Namibia, but concluded that considering the uncertainties associated with these estimates, the data would also be consistent with an active particle fraction of one. Thus, the increase of the microcline surface area present in the emulsion droplets by a factor of 100 only leads to an increase of the onset freezing temperature by 1 K. Such a slight increase of freezing temperature onsets with increasing suspension concentration was also observed in the case of the mineralogically quite pure kaolinite KGa-1b (Pinti et al., 2012) and is a sign of the presence of a characteristic active site on the surface that induces freezing in a narrow temperature range.

Atkinson et al. (2013) report an active site density of about $10^6$ cm$^{-2}$ for microcline between 251 K and 252 K. The freezing onset for 2 wt% microcline suspensions in pure water is ~252 K (see Fig. 2). We assume that for 2 wt% suspensions, droplets containing about 1000 particles are responsible for the freezing onset (see Sect. 3.1). This assumption results in an active site density of at least $5 \times 10^{-5}$ cm$^{-2}$ at 252 K. The same evaluation for 0.2 wt% and 20 wt% microcline emulsions yields active site densities of at least $5 \times 10^{-6}$ cm$^{-2}$ and at least $5 \times 10^{-4}$ cm$^{-2}$ at 251 K and 252 K, respectively. Note, that it is only possible to determine a lower limit

of active site densities because more than one site within a droplet may be responsible for freezing especially at high suspension concentrations. Considering the roughness of the estimate and that microcline samples from different sources and origins are compared, the active site densities of our emulsion freezing experiments are in excellent agreement with the ones reported in Atkinson et al. (2013).

Niedermeier et al. (2015) and Burkert-Kohn et al. (2017) performed IN experiments with single microcline particles immersed in a water droplet in chambers with residence times in the order of seconds. Niedermeier et al. (2015) observed onset freezing temperatures of 248 K. Their onset corresponds to a frozen fraction of 0.02 to 0.05 and does not vary much for particles with diameters from 200 nm to 500 nm. The 500 nm particles achieve an ice active fraction of one, while the smaller particles reach a plateau below an ice active fraction of one at 238 K. Burkert-Kohn et al. (2017) performed single particle immersion freezing experiments with size-segregated microcline and found a frozen fraction of 0.1 up to 253 K for 300 nm particles and up to 248 K for 200 nm particles. Overall, the freezing onsets of the single particle measurements seem to be slightly lower than the onsets of emulsion freezing experiments. This discrepancy may be explained at least partly by experimental differences: (i) the onset conditions of the single particle measurements corresponds with a higher frozen fraction of 0.02 – 0.05 compared with 0.001 estimated for the emulsion freezing experiments, (ii) the timescale of the single particle experiments was in seconds compared with minutes for emulsion freezing experiments, (iii) the microcline samples were not the same.

**4.2 Heterogeneous ice nucleation described by the water-activity-based approach**

The water-activity-based approach assumes that the decrease in freezing temperature with increasing solute concentration is solely a result of the freezing point depression with no or insufficient specific interactions with the particle surface to modify its IN efficiency and can therefore be described with $T_{\text{het}}^{\Delta a_w^{\text{het}}}(a_w) = T_{\text{melt}}(a_w + \Delta a_w^{\text{het}})$. Note that this derivation of the heterogeneous freezing temperatures on the basis of the solution water activity and the freezing temperature of the suspension in pure water is analogous to the one of ABIFM and results in the same predictions. If this approach were valid, the heterogeneous freezing temperatures of microcline solution droplets should align along the solid black line in Fig. 3, which was obtained with $\Delta a_w^{\text{het}} = 0.187$ needed to shift the ice-melting curve to the heterogeneous freezing temperature of the microcline suspension in pure water. The strong deviations of the heterogeneous freezing temperatures from this line imply specific interactions between microcline and the solutes. Different types of specific interactions are conceivable such as surface cation exchange, adsorption of solutes due to hydrogen bonding or van der Waals forces as well as irreversible surface destruction under extreme pH conditions. In the case of microcline, specific interactions are conceivable at different sites, namely at cationic and anionic centers. A cationic center is represented by $K^+$ while the polar silanol groups ($\equiv$Si-OH) or relatively non-polar siloxane groups ($\equiv$Si-O-Si$\equiv$) (Abramov, 1993) represent anionic centers.

**4.3 Increased IN efficiency at low solute concentration**

**4.3.1 Role of surface cation exchange**

When microcline particles are suspended in water, the native charge-balancing cations ($K^+$) from the surface cationic centers of microcline are exchanged by $H^+$ or rather by $H_3O^+$ (Fenter et al., 2003; Lee et al., 2008). This initial cation exchange is estimated to last approximately 1 min in pure water saturated with $CO_2$ and results in depletion in $K^+$ of the first layer at the surface (Busenberg and Clemency, 1976; Fenter et al., 2000; Chardon et al., 2006). The presence of additional external cations in a salt solution introduces competition in the cation exchange. The hydrated size of the externally introduced cations, the stability of

surface complexes and the pH of the solution are factors that govern the cation-exchange selectivity of the aluminosilicate surface (Garrels and Howard, 1959; Petrović et al., 1976; Auerbach et al., 2003; Ohlin et al., 2010; Belchinskaya et al., 2013). The externally added $NH_4^+$ (ionic radius of 1.42 Å) and $Na^+$ (ionic radius of 0.98 Å) ions can potentially exchange with the native $K^+$ (ionic radius of 1.33 Å) ions in competition with $H^+$.

Figure 6 shows $\Delta T_{het}$ (the difference between observed $T_{het}$ and expected $T_{het}$ depicted by the solid black line $T_{het}^{\Delta a_w^{het}}(a_w)$ in Fig. 3) vs. $[X^+]/[K^+]$ (the ratio of the concentration of externally added cations X (= $NH_4^+/Na^+/K^+$) to the concentration of surface $K^+$ available for the ion exchange; see Appendix A). The concentration of $NH_4^+$ in $NH_3$ solutions is based on the reversible equilibrium $NH_3 + H_3O^+ \leftrightarrow NH_4^+ + H_2O$. In the case of $NH_4^+$-containing solutes, $\Delta T_{het}$ increases up to 5.5 K as the cation ratio is increased to $[NH_4^+]/[K^+] \approx 1000$, while a further increase results in decreasing $\Delta T_{het}$. Interestingly, $\Delta T_{het}$ starts to increase only after $[NH_4^+]/[K^+]$ has reached a value of 1, with the exception of $(NH_4)_2SO_4$ solutions. Cation exchange on the surface is the primary process that takes place when $NH_4^+$ encounters microcline (Marshall, 1962; Demir et al., 2001; Demir et al., 2003). Since the affinity of the microcline surface towards $NH_4^+$ is very strong (Nash and Marshall, 1957), the surface ion replacement of $K^+$ by $NH_4^+$ can be considered absolute even at the lowest solute concentrations, implying that the ion exchange of $K^+$ by $NH_4^+$ is completed or almost completed when $[NH_4^+]/[K^+]$ reaches 1. Since the IN efficiency increases only after the ion exchange has been completed, the replacement of $K^+$ by $NH_4^+$ does not seem to be the reason for the enhanced IN efficiency in $NH_4^+$ containing solutions. Rather, the presence of adsorbed $NH_4^+$ or $NH_3$ on the surface seems to cause the improved ice nucleating capability of microcline (see Section 4.3.2). In the case of non-$NH_4^+$ solutes, a monotonic decrease in $\Delta T_{het}$ is observed as the exchangeable cation ratio increases. This is also the case for $K^+$ containing solutes which reduce the replacement of $K^+$ by $H^+$ implying that the particles will tend to retain surface-$K^+$.

Zolles et al. (2015) suggested that the superior IN efficiency of K-feldspars may be related to the kosmotropic (structure making) characteristic of $K^+$ in the surface layer of microcline. However, the experiments performed here do not confirm that the presence of $K^+$ in the microcline suspension has a positive effect on the IN efficiency. Rather, increasing the concentration of $K^+$ ions in the surface layer of the mineral by adding a $K^+$ containing solute decreases the IN efficiency compared to the pure water case. Also, the addition of $K_2SO_4$ has a similar deteriorating effect on the IN efficiency of microcline as the addition of $Na_2SO_4$ rendering an explanation of the superior IN efficiency of K-feldspars compared with Na-feldspars based on the nature of the charge neutralizing cation doubtful.

### 4.3.2 Role of adsorbed NH₃

A good lattice match is often listed among the requirements that may render a crystalline surface IN active (Pruppacher and Klett, 1994). However, Pedevilla et al. (2016) demonstrated with ab initio density functional theory calculations that ice-like overlayers do not form in the contact layer on the most easily cleaved (001) surface of microcline. Nevertheless, they concluded that this surface may induce ice-like ordering in the second overlayer by highlighting the role of surface hydroxyl (≡Al-OH/≡Si-OH) groups on the (001) plane of microcline. The –OH group may form hydrogen bonds with incoming water molecules on the surface resulting in a contact layer that may provide a template for further layers to arrange in ice-like manner. The –OH groups could also hold the answer to the variability in IN efficiency of various feldspars as they could be affected by weathering processes, chemical aging, microtexture, Al:Si order, etc. (Yang et al., 2014; Harrison et al., 2016; Peckhaus et al., 2016; Tang et al., 2016). Using molecular dynamics simulations, Lupi et al. (2014) and Lupi and Molinero (2014), have recently shown that even non-mineral graphitic surfaces can promote IN due to surface-induced ordered layering of interfacial water. Kiselev et al. (2016) suggested that

microscopic patches of the high energy (100) surface of feldspar should be responsible for the high IN efficacy of K-feldspars rather than the more common low energy (001) surface. If this were the case, the (100) face would need to be present on almost all particles since the majority of submicron microcline particles show IN activity (Niedermeier et al., 2015; Kaufmann et al., 2016). However, the (100) face is a high energy face that is not easily cleaved during milling and disappears during crystal growth. Moreover, Pedevilla et al. (2016) attribute IN activity to the common, easily cleaved (001) face, which we expect to dominate IN on submicron particles due to its prevalence on the particle surface. Whale et al. (2017) explain the exceptional ice-nucleating ability of alkali feldspars to microtextures related to phase separation into Na and K-rich regions. Since such features are very rare, they might account for the IN activity at the warmest temperatures observed in bulk experiments but are unlikely the reason for the ice active sites probed in our emulsion experiments, which are present in almost all submicron particles.

Several studies have shown that $NH_4^+$ is not only effective for surface cation exchange but also attaches to the aluminosilicate surface with high bonding energy (Nash and Marshall, 1957; Barker, 1964; Russell, 1965; Dontsova et al., 2005; Belchinskaya et al., 2013). The increase of $\Delta T_{het}$ for $[NH_4^+]/[K^+] < 1$ in case of the $(NH_4)_2SO_4$ and $NH_3$ solutions suggests $NH_3$ to be the interacting molecule, since these solutions contain $NH_3$ even at the lowest concentration. There is also an enhanced IN efficiency in terms of heterogeneously frozen fraction $F_{het}$ when microcline is suspended in $NH_3/NH_4^+$ containing solutions, as shown in Fig. 5. Since the degree of aggregation of microcline particles is not altered in dilute solutions compared to the pure water case (see Appendix B), the presence of $NH_4^+$ or $NH_3$ seems to activate nucleation sites that were inactive in pure water. The increased IN efficiency also observed in the case of dilute $NH_3$ solutions suggests that after the initial cation exchange, the aqueous $NH_3$ formed from excess $NH_4^+$ ions is adsorbed on the microcline surface and enhances its IN efficiency. $NH_3$ can make hydrogen bonds with surface hydroxyl groups with nitrogen atoms facing the surface and providing an abundant number of protons facing towards the bulk water allowing water molecules to orient with the protons facing the bulk (Wei et al., 2002; Anim-Danso et al., 2016). This orientation of hydrogen bonding provided by $NH_3$ may indeed be the reason for the enhanced IN efficiency of microcline in the presence of $NH_3$ and $NH_4^+$. The enhancement in IN efficiency is restricted to certain $NH_4^+$ concentrations as excess solute strength hampers the IN efficiency (discussed in Sect. 4.4). A positive effect of $NH_3$ on the IN efficiency in deposition mode was observed by Salam et al. (2007; 2008) when they exposed montmorillonite particles to $NH_3$ gas. Therefore, we suggest that $NH_3$ is able to improve the IN efficiency of active sites and even to create new active sites and thus to increase the overall number of active sites as indicated by the increase in $F_{het}$.

**4.4 Reduced ice nucleation activity at higher solute concentrations**

Strong reduction in the IN efficiency of size-segregated K-feldspar (76% microcline) particles has been reported after they were coated with sulfuric acid (Kulkarni et al., 2012; Augustin-Bauditz et al., 2014). It was found that coating thickness, hence concentration, and exposure time are important factors that govern the extent to which the acid treatment modifies the lattice structure of K-feldspar. This has been corroborated by Burkert-Kohn et al. (2017) who showed that the same feldspar (200 nm and 300 nm size segregated) turned almost ice inactive after treatment (12 hours) with 1 molar sulfuric acid and 1 molar nitric acid. Similar effects can be observed in the current study. Even the slightest increase in the sulfuric acid concentration (0 – 0.09 molal) drastically decreased $F_{het}$. Similarly, increasing the concentration of $NH_4HSO_4$ resulted in the reduction of $F_{het}$ while $T_{het}$ still showed an increase at low concentration ($a_w = 1 – 0.985$). We attribute these opposite effects to the competition between the enhancement of the IN efficiency by $NH_4^+$ and the destruction of IN active sites by $HSO_4^-$.

For all investigated salts, the IN efficiency decreases at higher solute concentration (Figs. 3 and 5). For a given water activity, $T_{het}$ and $F_{het}$ are highest for $NH_4Cl$ and decrease in the order of $NH_4Cl > NH_4NO_3 > NH_4HSO_4 > (NH_4)_2SO_4 > Na_2SO_4 > KCl > K_2SO_4$.

The difference in IN efficiency for salts with the same cation suggests that anions as well as cations play a role in the decrease of IN efficiency at higher concentrations.

The decrease in IN efficiency of microcline in suspensions with high solute concentrations shows two specific dependencies on the nature of the salt, namely, (i) the sulfate decreases the IN efficiency more strongly than chloride and nitrate, (ii) $K^+$ and $Na^+$ decreases the IN efficiency when compared with the pure water case and even more compared with $NH_4^+$.

The microcline surface has a negative charge at neutral pH arising from dissociated silanol groups (Demir et al., 2001; Demir et al., 2003; Karagüzel et al., 2005; Gülgönül et al., 2012). While at low concentrations, cations predominantly participate in ion

exchange, increasing the solute concentration leads to adsorption of cations on the negatively charged feldspar surface. This induces the buildup of an electrical double layer consisting of a compact inner layer occupied mainly by cations and a more diffuse outer layer made up of cations and anions. The size and charge of the ions influence the thickness of the layer and the zeta potential (Yukselen-Aksoy and Kaya, 2011). The double layer may influence the IN efficiency of microcline at higher solute concentration, since the high ion concentration at the particle surface may block nucleation sites.

The stronger decrease of IN efficiency in the presence of sulfate compared with other anions hints to specific interactions of sulfate with the IN active sites on the microcline surface. Indeed, sulfates have been described to attach strongly to feldspar surfaces (Priyantha and Perera, 2000). This adsorption has been explained by monodentate surface complexes of sulfate with tetrahedral Al sites (Min et al., 2015). The binding of sulfate leads to the release of $OH^-$, thus decreasing the number of Al-OH groups available for hydrogen bonding, resulting in a decrease of proton density on the microcline surface.

The addition of KCl decreases $F_{het}$ even at the lowest concentration of $7x10^{-5}$ molal. Increasing the $K^+$ concentration reduces the exchange of surface $K^+$ with $H^+$ and eventually prevents it. The total absence of IN activity at higher $K^+$ concentrations indicates that the replacement of $K^+$ with $H^+/H_3O^+$ is essential for the IN activity of microcline. This suggests that the higher IN efficiency of K-feldspars compared with (Na, Ca)-feldspars does not stem from the beneficial effect of $K^+$ in the surface layer. For instance, the comparison of $K_2SO_4$ with $Na_2SO_4$ solutions with the same water activity shows that the presence of $K^+$ has even a more

negative effect on $F_{het}$ than $Na^+$.

**4.5 Aging effect**

Suspension of feldspars in water leads to the immediate release of surface cations within minutes (Nash and Marshall, 1957; Busenberg and Clemency, 1976; Smith, 1994; Peckhaus et al., 2016). Over a period of years, the continuous release of silica as silicic acid, hydrated alumina and small structural fragments leads to the slow disintegration of feldspar and to the buildup of new

minerals (DeVore, 1957; Banfield and Eggleton, 1990). During this slow dissolution, a surface layer is considered to form on the feldspar surface (Busenberg and Clemency, 1976; Alekseyev et al., 1997; Zhu, 2005; Zhu and Lu, 2009). Interestingly, this slow dissolution of microcline is accompanied with only a slight reduction in IN efficiency even after several months as shown in experiments by Peckhaus et al. (2016) and Harrison et al. (2016). In order to assess the effect of solutes on IN efficiency over time, aging experiments were performed over a period of one week with microcline (2 wt%) suspended in ammonia (0.05 molal) and

ammonium sulfate (0.1 and 10 wt%) solutions as well as in pure water (Fig. 7). These are single measurements and uncertainties in $T_{het}$ and $F_{het}$ are assumed to be at maximum 0.8 K and 10%, respectively, taken from the data of multiple experiments performed on fresh microcline suspended in aqueous solution as shown in Fig. 3.

Figure 7 indicates that $T_{het}$ is preserved over seven days when microcline is suspended in pure water. However, $F_{het}$ dropped from 0.75 to 0.49 after one day and then remains constant within measurement uncertainties. Suspending microcline in dilute $NH_3$ and $(NH_4)_2SO_4$ solutions leads to an immediate increase in $T_{het}$ compared with the pure water case with no clear further trend over the next seven days. Suspending microcline in a concentrated $(NH_4)_2SO_4$ solution leads to an immediate decrease in $T_{het}$ and $F_{het}$ compared with the suspension in pure water with no clear further trend during the next days. This indicates, that in addition to the immediate effect of solutes on $T_{het}$ and $F_{het}$, dilute $NH_3$ and $(NH_4)_2SO_4$ solutions might have a long term preserving effect on the IN activity represented by $F_{het}$ remaining constant, maybe because of a stabilizing effect on the feldspar surface slowing down its dissolution in water.

The solution pH influences the surface charge of hydroxylated mineral surfaces. This again influences the ordering of water molecules at the water/mineral interface and the IN efficiency as was recently shown by Abdelmonem et al. (2017). In case of feldspars, the pH of the solution is, in addition, an important parameter in determining the stability of the feldspar surface. Extreme pH conditions enhance the dissolution of the aluminosilicate framework resulting in a faster surface degradation than under near-neutral conditions (Wollast, 1967; Busenberg and Clemency, 1976; Berner and Holdren, 1979; Skorina and Allanore, 2015). Indeed, deposition and immersion nucleation experiments showed reduced IN efficiency for sulfuric acid-coated feldspar particles (Kulkarni et al., 2014).

Given the dilute concentrations of $(NH_4)_2SO_4$ and $NH_3$ solutions used in the aging experiments, they provide only slightly acidic (pH 5.5) and basic conditions (pH 10.9), respectively. Therefore, the results suggest that even though the pH conditions play a role in the aging experiments, a counterbalancing stabilizing effect of $NH_3$ and $NH_4^+$ kept the IN efficiency of microcline more or less unaffected during the studied timescales.

**4.6 Reversibility of surface modifications**

From the above sections we understood how certain inorganic species affect the IN efficiency of microcline. To investigate the reversibility of these modifications we ran emulsion freezing experiments on microcline samples that were first aged under concentrated solute conditions for 10 days and then resuspended in pure water or a dilute solution. The samples were aged under the following conditions: 10 wt % $(NH_4)_2SO_4$ (pH 5.5), 2 wt % $NH_4HSO_4$ (pH 1.2), 0.5 wt % $K_2SO_4$ (pH 8.4), and 2 molal $NH_3$ (pH 11.7) solutions and then resuspended in pure water and dilute solution. The aim of the reversibility tests are twofold, namely (i) to learn more about the properties of active sites by testing their stability under severe pH conditions, (ii) to investigate the relevance of microcline as an INP after atmospheric aging.

The upper panel in Fig. 8 shows the change in $T_{het}$ between the treated particles (resuspended in water or dilute solutes) and its fresh counterpart ($\Delta T_{het} = T_{het, treated} - T_{het, fresh}$). The lower panel in Fig. 8 shows the relative decrease in $F_{het}$ for the treated particles with respect to its fresh counterpart, expressed as fraction (($\Delta F_{het})_{relative} = (F_{het, treated} - F_{het, fresh})/ F_{het, fresh}$). $\Delta T_{het}$ and ($\Delta F_{het})_{relative}$ are represented as a function of the pH of solution used for aging over 10 days. We chose a presentation as a function of pH because we consider the increased dissolution rates at low and high pH (see Sect. 4.5) as a determinant of IN activity after aging because it enhances the degradation of the surface which may result in the irreversible loss of active sites.

Although the decrease in $T_{het}$ in a freshly prepared 2 molal $NH_3$ solution is in the range expected according to the water-activity-based approach (as shown in Fig. 3), there is no recovery when the microcline is resuspended in pure water or in a dilute $NH_3$ solution (0.05 molal; pH 10.2) after ten days. The irreversible loss of IN activity is even clearer in terms of ($\Delta F_{het})_{relative}$. Similarly, although $T_{het}$ is even increased when a microcline suspension freshly prepared in 2 wt% $NH_4HSO_4$ is frozen, the acidic conditions

in the suspension lead to the irreversible loss of IN activity after ten days as can be seen from $\Delta T_{het}$ = -6 to -7 K and only a partial recovery in terms of $F_{het}$ of the resuspended samples in pure water and in a dilute $NH_4HSO_4$ (0.5 wt%; pH 1.6) solution.

On the other hand, the loss of IN efficiency in concentrated $(NH_4)_2SO_4$ (10 wt%) and $K_2SO_4$ (0.5 wt%) is largely reversible. While $T_{het}$ is almost fully recovered, $F_{het}$ of the resuspended samples remains below the value of the freshly prepared suspensions. This shows that aging at near-neutral conditions leads to less permanent damage of the microcline surface than aging under more extreme pH conditions. Rather, in the presence of $(NH_4)_2SO_4$ and $K_2SO_4$ a part of the nucleation sites seem to be reversibly shutdown by the solutes and are able to recover.

The dependence of the recovery of IN activity on the solution pH during aging, suggests that the irreversible loss of IN activity is related to the pH dependence of the dissolution rate of microcline. The dissolution of feldspars is slowest at neutral or near neutral conditions (pH 3 – 8) and increases towards low and high pH (Helgeson et al., 1984). At low pH the cations and Al are depleted, and an amorphous Si-enriched surface layer may form (Chardon et al., 2006; Lee et al., 2008). Dissolution is supposed to occur predominantly at high energy sites, such as kink sites or defects, leading to a roughening of flat areas on freshly cleaved surfaces and to rounding of the steps (Chardon et al., 2006). These surface modifications seem to correlate with the irreversible loss of IN active sites that we observe at acidic (pH ≲ 1.2) and alkaline (pH ≳ 11.7) conditions.

## 5 Atmospheric implications

Mineral dust particles, when lifted into the upper troposphere, have lifetimes of several days and can be transported over long distances (Huneeus et al., 2011). Quartz is the dominant component of dusts collected near the source region, while the clay mineral fraction dominates at locations far away from the source (Murray et al., 2012). Nevertheless the minor contributions of feldspars to atmospheric dust aerosols have been suggested to determine the IN efficiency of mineral dusts under mixed-phase cloud conditions because of their high IN efficiency (Atkinson et al., 2013; Tang et al., 2016). The temperature regime probed in our experiments is relevant to mixed phase clouds (273 K - 236 K) where liquid phase is observed before ice crystal formation (Ansmann et al., 2009; Wiacek et al., 2010; de Boer et al., 2011).

Transported mineral dust particles may acquire a coating when they come in contact with reactive gases and semivolatile species (Usher et al., 2003; Kolb et al., 2010; Ma et al., 2012; He et al., 2014; Tang et al., 2016) or when they undergo cloud processing (Fitzgerald et al., 2015). The chemical composition and thickness of the coating depend on the particle mineralogy and on the transport pathway (Matsuki et al., 2005a; Sullivan et al., 2007; Fitzgerald et al., 2015). Calcium-rich particles, originating from calcite, have been shown to be more susceptible to processing by $SO_2$ and $NO_y$ than silicates and aluminosilicates (Matsuki et al., 2005b). Furthermore, airborne and ground station measurements imply that Saharan dust particles undergo little chemical processing during long-range transport across the Atlantic unless they become incorporated in cloud droplets by acting as cloud condensation nuclei (CCN) (Matsuki et al., 2010; Denjean et al., 2015; Fitzgerald et al., 2015). An exception includes desert dusts originating from the industrial regions at the Atlantic coast of Morocco, Algeria, and Tunisia, which show high concentrations of ammonium, sulfate, and nitrate when sampled at Tenerife, Canary Islands (Rodríguez et al., 2011). Conversely, the exposure of Asian dust to more polluted air masses with higher concentrations of reactive gases shifts its population towards a prevalence of coated particles even in the absence of cloud processing. By means of online single-particle mass spectrometry onboard a research vessel, Sullivan et al. (2007) found that high amounts of sulfate accumulated on aluminosilicate-rich Asian dust, while calcium-rich particles became enriched in nitrate. When passing through polluted regions with agricultural activity, the sulfates and nitrates may become neutralized by ammonia (Sullivan et al., 2007). By analyzing satellite data, Ginoux et al. (2012) found that soil dust

originating from cropland is often mixed with ammonium salts already before long-range transport. Since particles with soluble coatings are more susceptible to cloud droplet activation, they have an enhanced potential to induce cloud droplet freezing in immersion or condensation mode than uncoated particles.

The emerging picture suggests to discriminate situations when dust in the atmosphere is either mainly exposed to rather low $NH_3$ conditions, which keeps the aerosols acidic, or to polluted, highly ammoniated situations. We discuss the fate of microcline in atmospheric solution droplets following atmospheric air parcel trajectories P1 through P3 (red and yellow arrows) with increasing moisture (as shown in Fig. 9). As the basis of this discussion we use the freezing onsets and heterogeneously frozen fractions shown in Figs. 3 and 5, respectively. Figure 9 therefore shows freezing onsets that reflect the IN activity of the best microcline

particle out of about 1000. Scenarios for average microcline particles would be exactly the same, only with all reported temperatures shifted downwards to about the maximum of the heterogeneous freezing signal, i.e. by 2 – 4 K. The phase diagrams show $T$ vs $a_w$ in Fig. 9A and the supersaturation with respect to ice, $S_{ice}$ vs $T$ in Fig. 9B. The three atmospheric air parcels P1, P2, and P3 contain 0.3 hPa, 1.5 hPa and 2.0 hPa $H_2O$ partial pressure, respectively, and are supposed to cool adiabatically, attaining ice saturation at 242 K, 257 K and 260 K (or water saturation at 238.8 K, 255.1 K and 258.4 K, respectively). To stay in equilibrium

with the increasing relative humidity, the solution droplets take up water so that their water activity corresponds to the relative humidity in the air parcel. Subsequently, we distinguish ammonia-rich solutions, represented by $(NH_4)_2SO_4$, and highly acidic solutions, represented by $H_2SO_4$. Fig. 9 shows the corresponding heterogeneous freezing lines, color-coded with respect to the heterogeneously frozen fraction $F_{het}$ (see color bar), according to Fig. 3 for heterogeneous onset temperatures $T_{het}$ and Fig. 5 for the heterogeneously frozen fraction $F_{het}$ of microcline,.

*Parcel P1 – dry conditions.* Under dry conditions, microcline-containing $(NH_4)_2SO_4$-$H_2O$ droplets may nucleate ice heterogeneously in immersion mode (marked "imm" in Fig. 9) around 239 K ($a_w \approx 0.97$ $S_{ice} \approx 1.35$), albeit only with a reduced $F_{het}$. In contrast, microcline particles in $H_2SO_4$-$H_2O$ solution droplets have likely lost their entire IN efficiency. Upon continued adiabatic cooling, these particles follow the yellow arrows, form supercooled water cloud drops (filled blue circle), and eventually nucleate ice homogeneously ("hom") about 3 K lower than in the ammoniated case, namely around 236 K ($a_w = 1$, $S_{ice} \approx 1.42$).

*Parcel P2 – moist conditions.* These intermediate conditions can take advantage of the ammonia-induced enhancement of the IN efficiency of microcline. The $(NH_4)_2SO_4$-$H_2O$-containing microcline particles may nucleate ice heterogeneously at temperatures as high as 256 K ($a_w \approx 0.995$, $S_{ice} \approx 1.16$). We refer to this scenario as condensation freezing, although it is strictly speaking immersion freezing in a dilute solution droplet. However, given that nucleation occurs at $a_w = $ RH $\approx 99.5\%$, in experimental settings, freezing would appear to occur concomitantly with droplet activation, thus fulfilling the definition for condensation freezing

(marked "cond" in Fig. 9). In contrast, microcline particles in $H_2SO_4$-$H_2O$ solution droplets presumably have lost their nucleation efficacy irreversibly and continue to follow the yellow arrows upon further cooling and form supercooled cloud drops. Ice nucleates again only homogeneously (at "hom" in Fig. 9). If the aerosols in P2 are not ammoniated but acidic, the low pH will largely deactivate microcline, so that the solution droplets continue to follow the yellow arrows upon further cooling, forming supercooled liquid clouds and finally ice through homogeneous nucleation (at "hom" in Fig. 9).

*Parcel P3 – wet conditions.* Though wetter than P2, even in the presence of microcline in $(NH_4)_2SO_4$-$H_2O$ droplets, ice nucleates only after the air parcel formed a supercooled liquid water cloud, which eventually glaciates upon further cooling at the freezing onset temperature of microcline in pure water ($T_{het} \approx 252$ K, $a_w = 1$, $S_{ice} \approx 1.23$). This leads to the formation of a mixed phase cloud via immersion freezing ("imm" in Fig. 9). However, when drier air is mixed into the liquid cloud before the onset temperature for immersion freezing of microcline in pure water is reached, e.g. caused by entrainment/detrainment processes across cloud edges,

the droplets might move into the zone of enhanced microcline IN efficacy, thus triggering ice formation at several degrees higher temperature. Figure 9 shows this "mixing nucleation" process by a green double arrow (termed "mix"). Of course, it will be experimentally difficult to distinguish "mix" from "imm" or "cond". Finally, if the aerosols in P3 are not ammoniated but acidic, the low pH will largely deactivate microcline, so that the solution droplets continue to follow the yellow arrows upon further cooling, forming supercooled liquid clouds and finally ice through homogeneous nucleation (at "hom" in Fig. 9).

The "mixing nucleation" affecting P2 ("mix" in Fig. 9) is a process that enables microcline to trigger ice formation at $T_{het} \lesssim 256$ K, i.e. up to 4 K higher temperature than ice nucleation on microcline in pure water at $T_{het} \lesssim 252$ K. Interestingly, nucleation is triggered by drying an air parcel of a mixed-phase cloud temporarily and marginally to $a_w \lesssim 0.998$. Indeed, an enhancement of IN in evaporating clouds has been observed in field measurements and was referred to as evaporation freezing (Hobbs and Rangno, 1985; Beard, 1992; Cotton and Field, 2002; Ansmann et al., 2005; Baker and Lawson, 2006). Different causes of such evaporation
freezing have been put forward. On one hand, a drying would be expected to make IN less likely, because evaporation of water decreases the water activity. On the other hand, it causes evaporative cooling (Shaw and Lamb, 1999; Satoh et al., 2002) resulting in freezing, if temperatures fall below the homogeneous nucleation temperature briefly before the droplet fully evaporates or solutes become too concentrated. However, under atmospheric conditions the temperature difference between a droplet and its surroundings is typically smaller than 1 K (Neiburger and Chien, 1960), rendering this explanation implausible. Another process
that has been invoked is the emergence of so-called "evaporation ice nuclei" from a small fraction of ice particle residues (Kassander et al., 1957; Beard, 1992). Furthermore, enhanced contact freezing due to thermophoretic capture of submicron particles by evaporating droplets (Hobbs and Rangno, 1985; Beard, 1992) and contact freezing inside-out have been put forward as explanations for evaporation nucleation (Durant and Shaw, 2005; Shaw et al., 2005). "Mixing nucleation", as described here, may be a plausible alternative freezing process in case of ammoniated mineral dust particles.

The idealized trajectories P1 – P3 in Fig. 9 exemplify that in a condensation-freezing or mixing-nucleation scenario a higher freezing temperature may be achieved than for immersion freezing. The case of microcline demonstrates that in addition to the mineralogy, the chemical exposure history of the particles is a relevant determinant of the IN efficiency of airborne dust. A coating by an aqueous ammonium sulfate solution can indeed enhance the IN efficiency of microcline in the condensation mode, but only when sulfuric acid and ammonia are deposited concomitantly. Such a scenario may arise when the dust originates from
anthropogenically influenced regions with agricultural activity, as may be the case for soil dust (as described in Ginoux et al. (2012)). Indeed, IN measurements of air masses advected from the Sahara to the Canary Islands showed that ammonium sulfate, linked to anthropogenic emissions in upwind distant anthropogenic sources, mixed with desert dust had a small positive effect on the condensation mode INP per dust mass ratio but no effect on the deposition mode INP (Boose et al., 2016). If the microcline-containing aerosols acquire a sulfuric acid coating preceding the neutralization by ammonia, the IN activity may be destroyed
irreversibly. In the situation described by Sullivan et al. (2007), the mineral dust-laden air masses were first influenced by a volcanic eruption, likely with high $H_2SO_4$ concentrations, before they passed through polluted air masses rich in ammonia. Here, the neutralization might have occurred too late to preserve the IN active sites.

**6 Conclusions and Outlook**

Immersion freezing experiments with a DSC on microcline suspended in solutions containing various inorganic solutes of different
concentrations showed that the heterogeneous freezing onset temperatures deviate from the previously established water-activity-based approach. An increase in IN efficiency was observed in dilute solutions of $NH_3$ (< 1 molal) and $NH_4^+$-containing salts (<

0.16 molal) while a strong decrease was observed in aqueous microcline suspensions containing solutes with cations other than $NH_4^+$.

Neither the native surface $K^+$ ions nor their exchange with externally added cations were found to be the reason behind microcline being so highly IN efficient. The observed increase in IN efficiency in dilute $NH_3$ and $NH_4^+$-containing solutions seems to be related with chemically adsorbed ammonia molecules on the particle surface. Hydrogen bonded $NH_3$ molecules on the microcline surface might provide a better orientation of water molecules into ice-like layers via multiple protons available for hydrogen bonding.

Aging experiments over several days reveal solute-specific impacts on the IN efficiency of the microcline surface. In acidic (pH ≲ 1.2) or alkaline (pH ≳ 11.7) conditions, the loss of IN activity is irreversible. In contrast, the microcline surface is able to partially regain its IN efficiency when the particles are resuspended in water after aging in concentrated solutions with near-neutral pH (5.5 – 8.4).

The increased IN efficiency in dilute ammonia containing microcline droplets opens up a pathway for condensation freezing occurring at a warmer temperature than immersion freezing. Ammonia-rich conditions are expected in polluted or agricultural areas. Conversely, the IN efficiency of microcline is permanently destroyed when microcline acquires a sulfuric acid coating while passing through more pristine areas.

**7 Data availability**

The data for freshly prepared microcline suspensions in water or aqueous solutions (Fig. 3 and 5), aging tests (Fig. 7) and particle size distributions of suspensions obtained from the laser diffraction particle size analyser (Fig. B1) presented in this publication are available at the following DOI: 10.3929/ethz-b-000229892.

**Appendix A: Surface cation exchange**

Figure 6 shows $\Delta T_{het}$ (difference in observed $T_{het}$ and $T_{het}^{\Delta a_w^{het}}(a_w)$) as function of the ratio of externally added cations (from the solute) to native potassium ions $K^+$ available for exchange on the microcline surface. The externally added solutes dissociate into cations and anions in solution. The number of cations ($N_{cation}$) externally added from the solute is given as:

$$N_{cation} = moles_{cation} \, N_A \tag{2}$$

where $moles_{cations}$ is the moles of cations added to the solution and $N_A$ is the Avogadro constant ($6.023 \times 10^{23}$ per mole).

Figure A1 shows a simplified parallelepiped version of a primitive unit cell of microcline (chemical composition $K_4Al_4Si_{12}O_{32}$). It also shows the two most stable cleavage planes (001) AB and (010) AC. Surface $K^+$ can be released from the surface planes when a crystal is submerged in water. We assume that all $K^+$ from half of the unit cell are available for exchange, hence only the $K^+$ nearest to a plane can potentially move out from that particular plane. Assuming a microcline crystal with the morphology reflecting the unit cell, the fraction of surface area contributed by the different planes is given by:

$$f_{ij} = \frac{a_{ij}}{\Sigma_{ij=A,B,C} \, a_{ij}} \qquad i \neq j, \tag{3}$$

where $a_{ij}$ is the surface area of the different planes. The surface density of $K^+$ contributed by each plane can be calculated as:

$$\sigma_{ij} = \frac{n_{ij}}{a_{ij}} \qquad i \neq j \tag{4}$$

Where $n_{ij}$ is the number of exchangeable $K^+$ in the plane $ij$, with values of $n_{AB} = n_{BC} = 2$, $n_{AC} = 1$. Thus, the average surface density of exchangeable $K^+$ for a microcline crytstal is:

$$D = \sum_{ij=A,B,C} f_{ij}\sigma_{ij} \qquad i \neq j \tag{5}$$

The total number of $K^+$ released per unit mass of our microcline sample is the product of its BET surface area ($1.91 \, m^2 \, g^{-1}$) and the average surface density of $K^+$ (D). We assume that each surface cationic center is occupied by $K^+$ before it is suspended. The number of native potassium ions in Figure 6 is then the product of D and the mass of microcline added to the suspension. The error bars in $\Delta T_{het}$ are the maximum and minimum difference in observed $T_{het}$ and $T_{het}^{\Delta a_w^{het}}(a_w)$. The error bars in cation ratio stems from the maximum and minimum surface density (D) corresponding to planes AC and BC, assuming that the whole surface is made up of single type of plane.

### Appendix B: Aggregation of microcline particles

Depending on the surface charge, mineral dust may aggregate and coagulate when suspended in solution. Aggregation and coagulation may reduce the surface area available for IN and decrease the heterogeneously frozen fraction (Emersic et al., 2015). The dissociation of silanol groups is the primary factor governing the surface charge of feldspar resulting in a negative zeta potential in pure water (Demir et al., 2003). The addition of solutes can influence the surface charge either by changing the pH of the solution or by direct interaction with the surface.

To check whether aggregation is responsible for the dependence of $F_{het}$ on the solutes present in the suspensions, we determined the particle size distribution of microcline in pure water, 0.02 wt% $NH_4Cl$, 2 wt% $NH_4Cl$ and 0.5 wt% KCl with the Beckman Coulter LS13 320 laser diffraction particle size analyser (5 mW laser diode, wavelength 780 nm; coupled with Polarization Intensity Differential Scattering (PIDS) assembly). Particle size distributions were obtained for microcline freshly suspended and for the same samples after aging them for 2 hours in suspension (typical emulsion freezing measurement time span with the DSC). The freshly prepared suspensions were sonicated for 2 minutes before the size distribution measurement. Each suspension was measured three times and averages are reported.

Figure B1 shows the particle size distributions for fresh and aged suspensions. The addition of the solutes leads to a slight shift of the size distribution to larger sizes compared with the pure water case. Pure water and 0.02 wt% $NH_4Cl$ suspensions show negligible agglomeration during aging for 2 hours. Strong aggregation during aging for 2 hours was observed for 0.5 wt% KCl which may be explained by the enhanced neutralization of the surface charge due to the common ion.

Based on these results, the decrease of $F_{het}$ in the presence of KCl can be partly explained by aggregation, while the increase of $F_{het}$ in the presence of $NH_4Cl$ cannot be explained by a decrease of agglomeration in the presence of the solute. We therefore conclude that the increase of $F_{het}$ in the presence of ammonia containing solutes can be ascribed to the activation of sites due to the interaction of ammonia with the microcline surface.

*Acknowledgements.* This work was supported by the Swiss National Foundation, project number 200020_156251. We thank the following colleagues from ETH Zürich: Dr. Michael Plötze, Annette Röthlisberger, and Marion Rothaupt for the possibility to do XRD and BET measurements; Robert David for providing the SMPS and the APS and the strong support during size distribution

measurements; Peter Brack for providing various minerals. We also thank Silvan von Arx from the Institute of Mechanical Engineering and Energy Technology (Lucerne School of Engineering and Architecture, Lucerne) for size distribution measurement with the Beckman Coulter Laser Diffraction Particle Sizer.

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

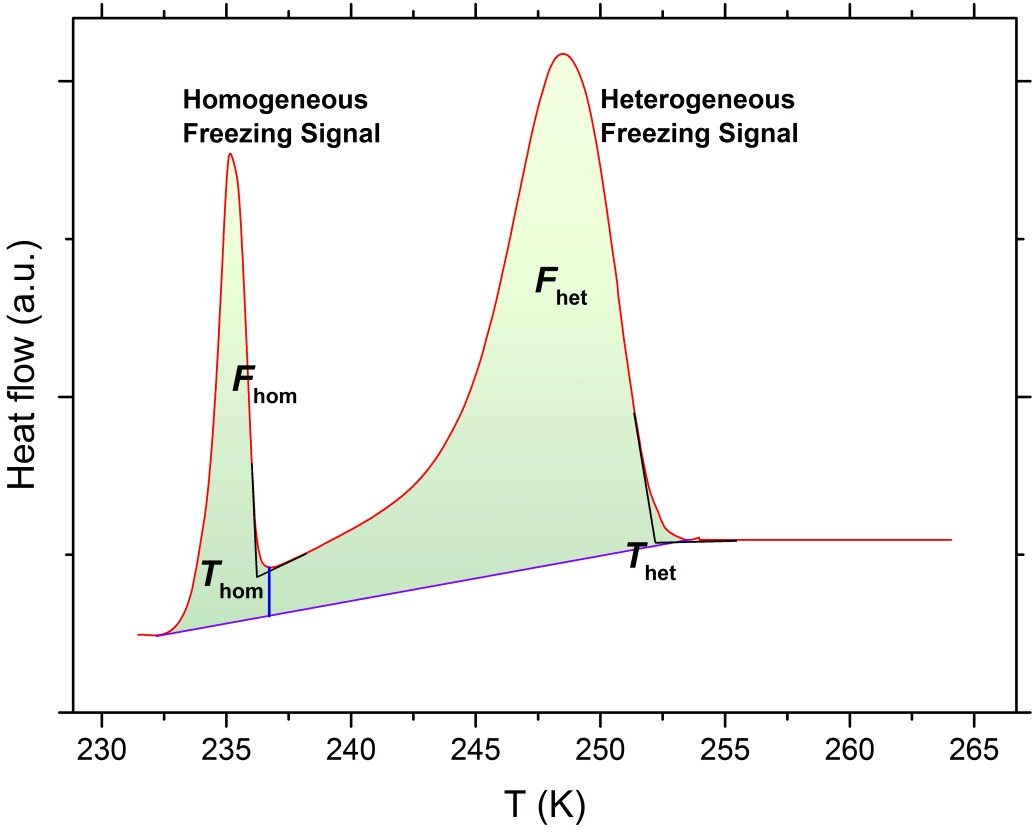

Figure 1. A typical DSC thermogram, showing freezing onset temperatures $T_{het}$ and $T_{hom}$ and the asymptotes used for their construction (black lines), as well as frozen fractions $F_{het}$ and $F_{hom}$ (shaded areas), which are normalized $F_{het} + F_{hom} = 1$. The area under each peak corresponds to the volume of water that froze homogeneously or heterogeneously. The straight violet line connects the heterogeneous freezing signal onset with the end of the homogeneous freezing signal) and is taken as the base line for evaluating the total frozen fraction. The vertical blue line marks the minimum intensity between the homogeneous and heterogeneous freezing peak and is taken as the separator between the areas under the heterogeneous and homogeneous freezing peaks. Note that frozen fractions are determined in the time domain (heat flow as a function of time) and not in the temperature domain shown here for illustration.

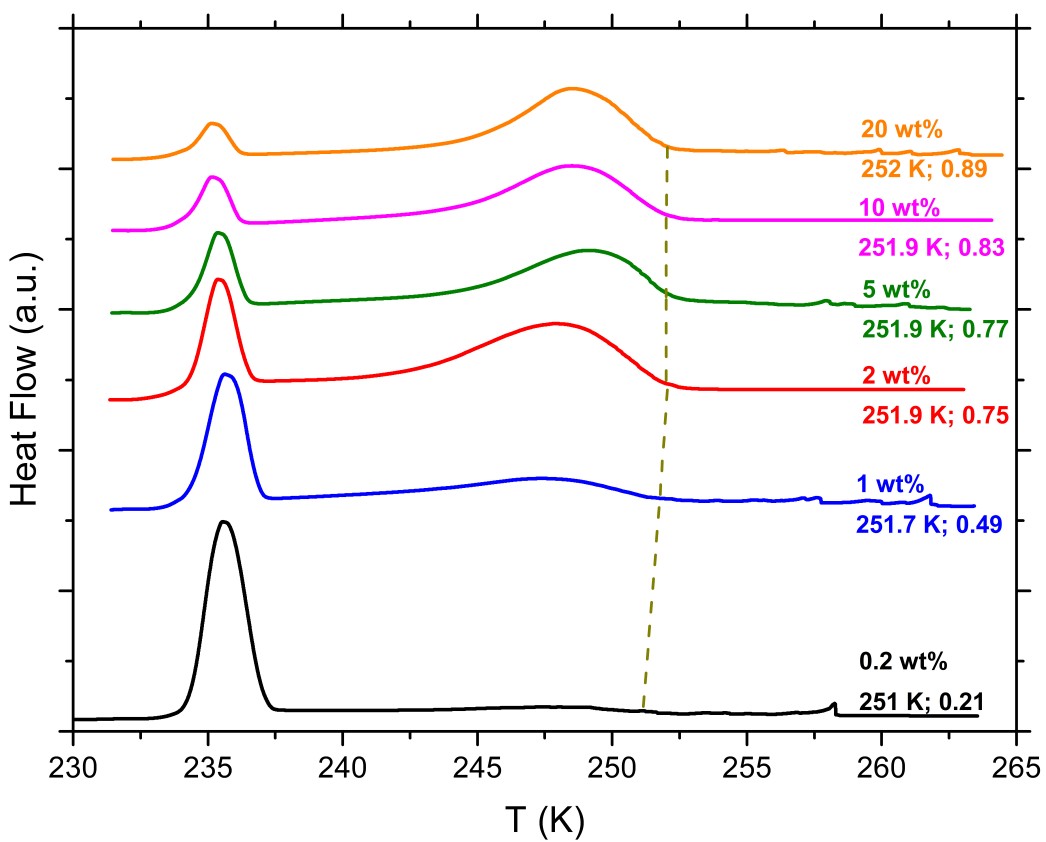

Figure 2.  DSC thermograms of varying suspension concentrations of microcline particles in pure water. All curves are normalized such that the areas under the heterogeneous and homogeneous freezing curves sum up to the same value. Numbers next to each curve: dust concentration (in wt%), onset temperature $T_{het}$ (in K) and heterogeneously frozen fraction $F_{het}$ (ratio of heterogeneous to total freezing signals, dimensionless), respectively. Note that the spikes at $T > 255$ K are due to the freezing of single particularly large emulsion droplets. They are not reproducible and therefore excluded from evaluation.

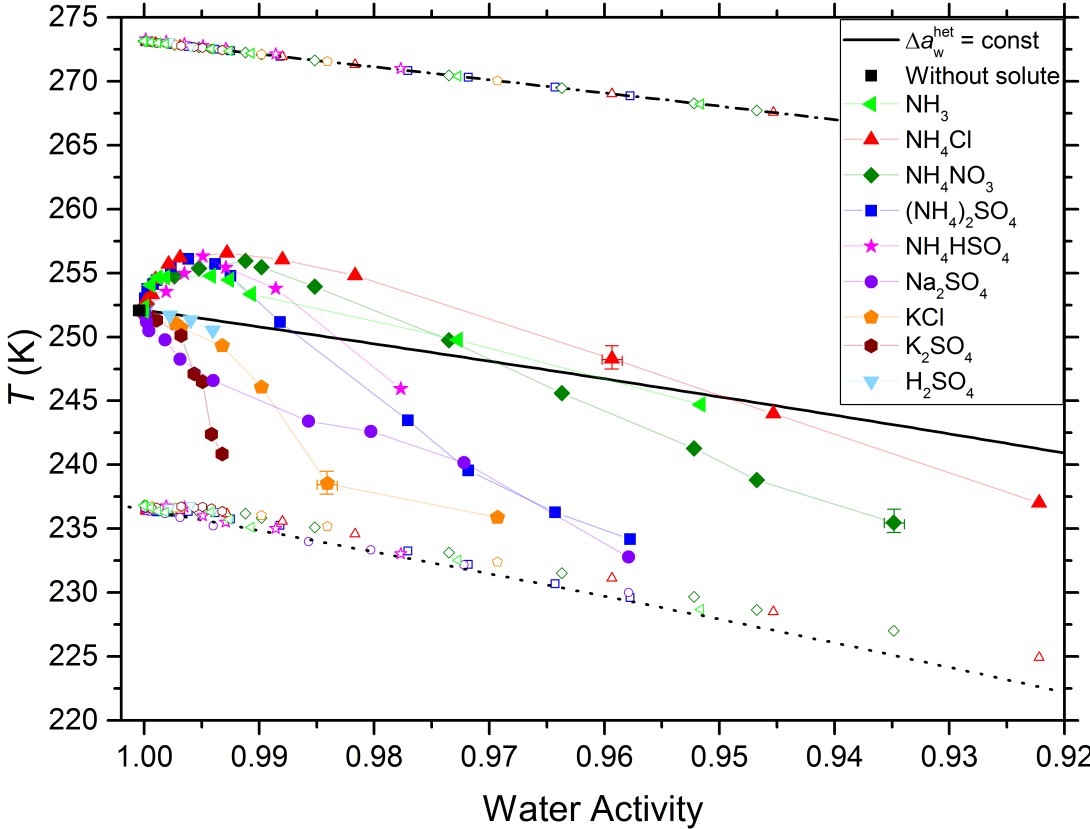

Figure 3. Compiled results of the freezing experiments with microcline. Heterogeneous freezing onset temperatures, $T_{het}$ (filled solid symbols connected via thin lines), and homogeneous freezing onset temperatures, $T_{hom}$ (open symbols at $T = 225 – 237$ K), and ice melting temperatures, $T_{melt}$ (open symbols at $T = 267 – 273$ K) as functions of the solution water activity, $a_w$, for various solutes (symbols and colors see insert). All suspensions contain 2 wt% microcline. Dash-dotted black line: ice melting point curve. Dotted black line: homogeneous ice freezing curve for supercooled aqueous solutions obtained by horizontally shifting the ice melting curve by a constant offset $\Delta a_w^{hom}(T) = 0.296$. Solid black line: horizontally shifted from the ice melting curve by $\Delta a_w^{het}(T) = 0.187$ derived from the heterogeneous freezing temperature of the suspension of microcline in pure water (filled black square at $a_w = 1$). Symbols are the mean of at least two separate emulsion freezing experiments. Three symbols carry error bars to show representative experimental variations (min-to-max) in $T_{het}$ and $a_w$.

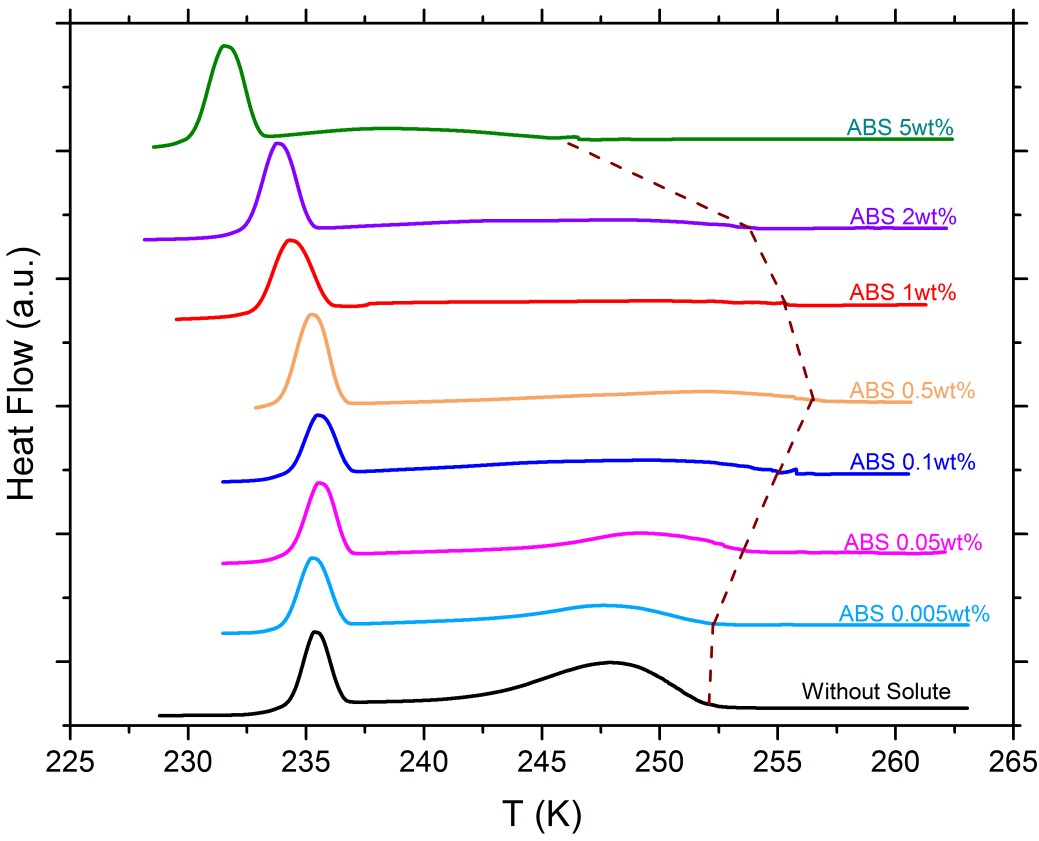

**50**

Figure 4. DSC thermograms of 2 wt% of microcline particles suspended in ammonium bisulfate (ABS) solution droplets of varying concentration (0-5 wt% ABS). All curves are normalized such that the areas under the heterogeneous and homogeneous freezing curves sum up to the same value. The dashed line connects the heterogeneous freezing onset temperatures ($T_{het}$) of the emulsions. With increasing ABS concentration $T_{het}$ increases initially, then decreases sharply. In contrast, the intensity of the

**55**    heterogeneous freezing signal decreases monotonically, implying continuous decrease in $F_{het}$.

**60**

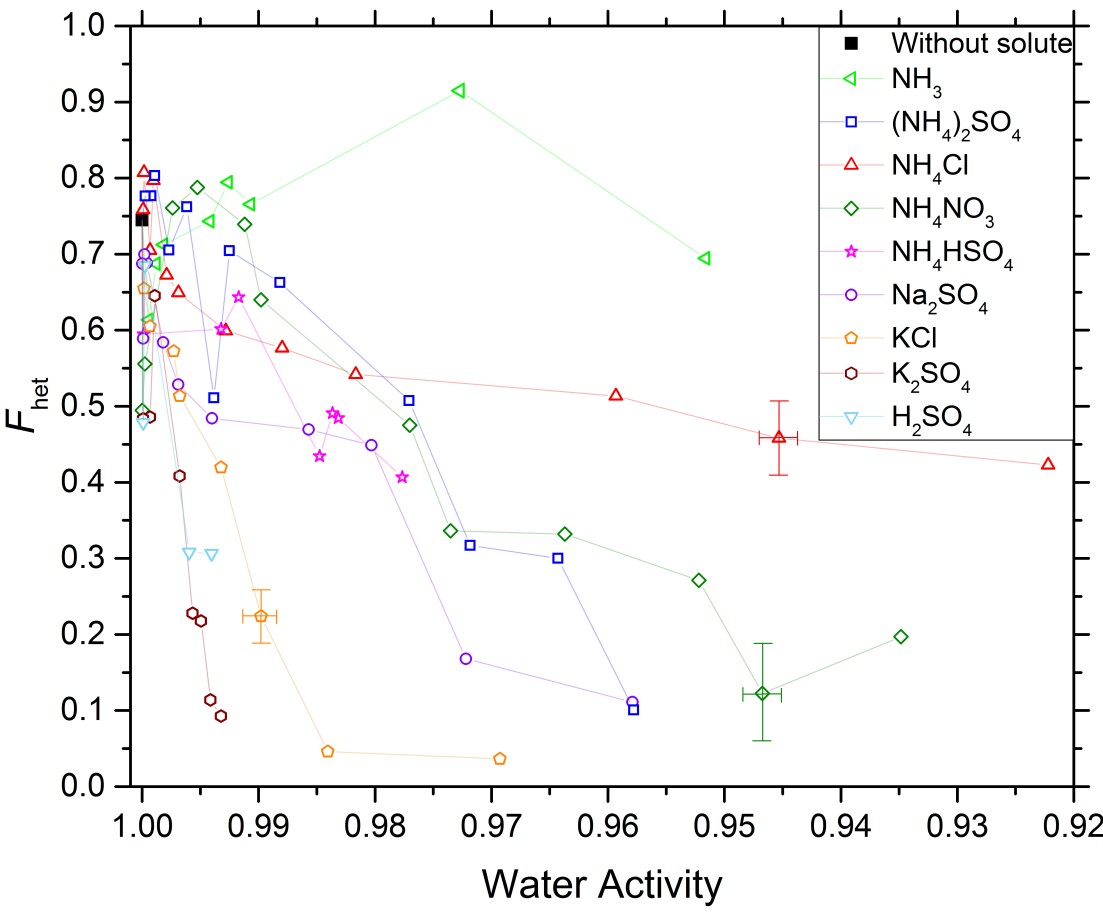

Figure 5. $F_{het}$ (volume fraction of water frozen heterogeneously, see Fig. 1) as function of solution water activity ($a_w$). All suspensions contain 2 wt% microcline. Three symbols carry error bars showing representative experimental variations (min-to-max) in $F_{het}$ and $a_w$. Absolute uncertainties in $F_{het}$ do not exceed ± 0.1.

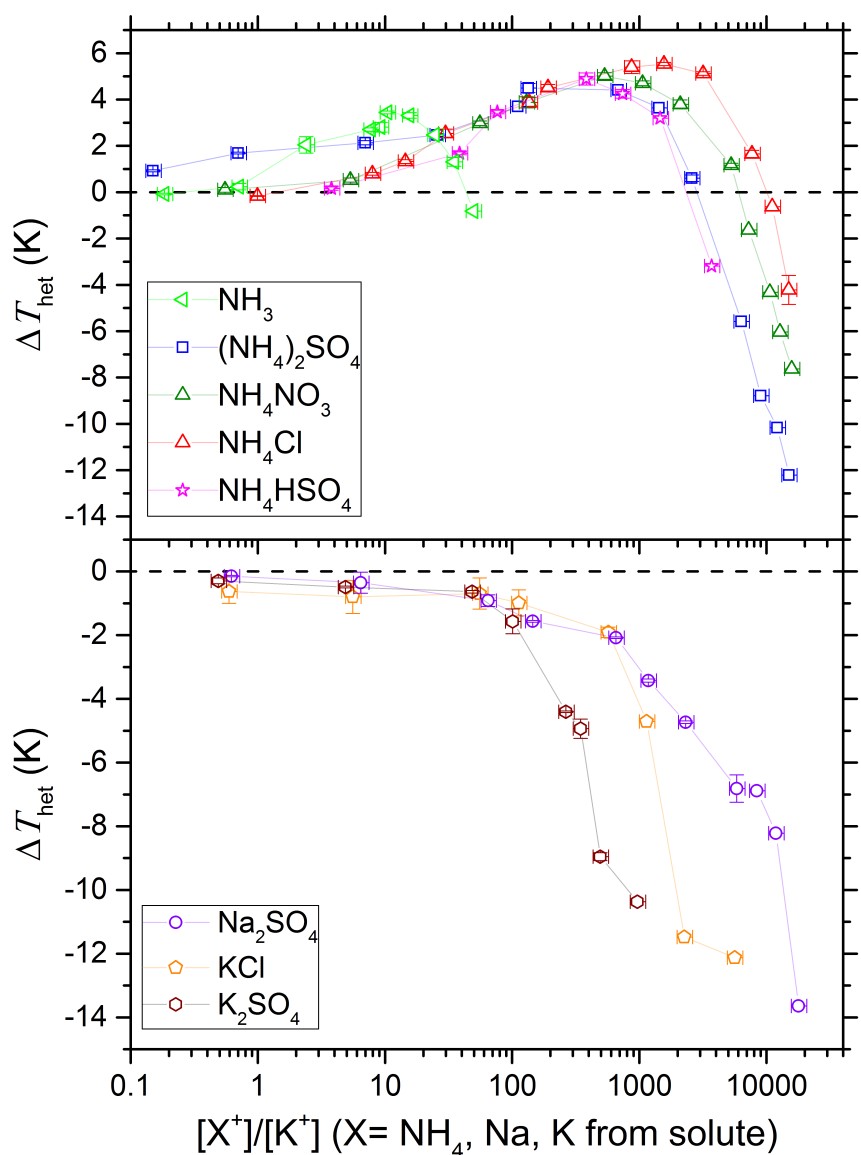

Figure 6. $\Delta T_{het}$ (difference in observed $T_{het}$ and $T_{het}^{\Delta a_w^{het}}(a_w)$) as a function of ratio of number of externally added cations (from solute) to native potassium ions, $K^+$, available for exchange on the microcline surface. Upper panel: $NH_3$ and $NH_4^+$ containing solutes. Lower panel: non-$NH_4^+$ solutes. The dashed black lines in both panels depict $\Delta T_{het} = 0$ (shown to guide the eye).

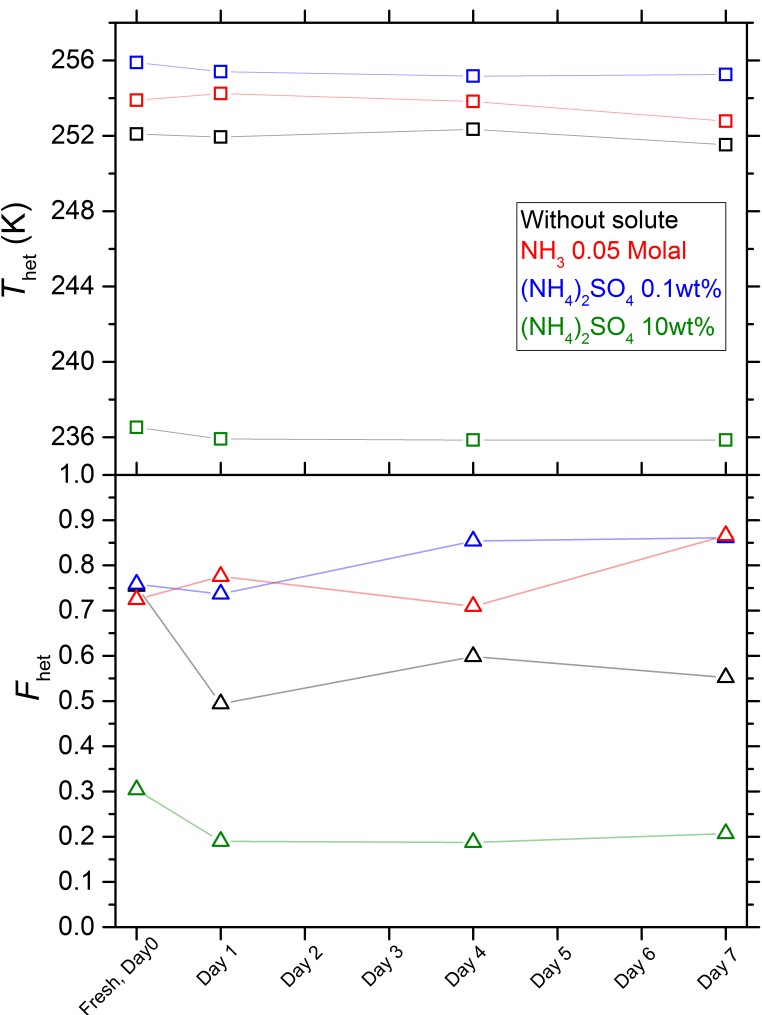

Figure 7. Development of $T_{het}$ (upper panel - squares) and $F_{het}$ (lower panel - triangles) for 2 wt% microcline suspended in water, 10wt % $(NH_4)_2SO_4$, 0.1wt % $(NH_4)_2SO_4$, 0.05 molal ammonia solutions over a period of one week.

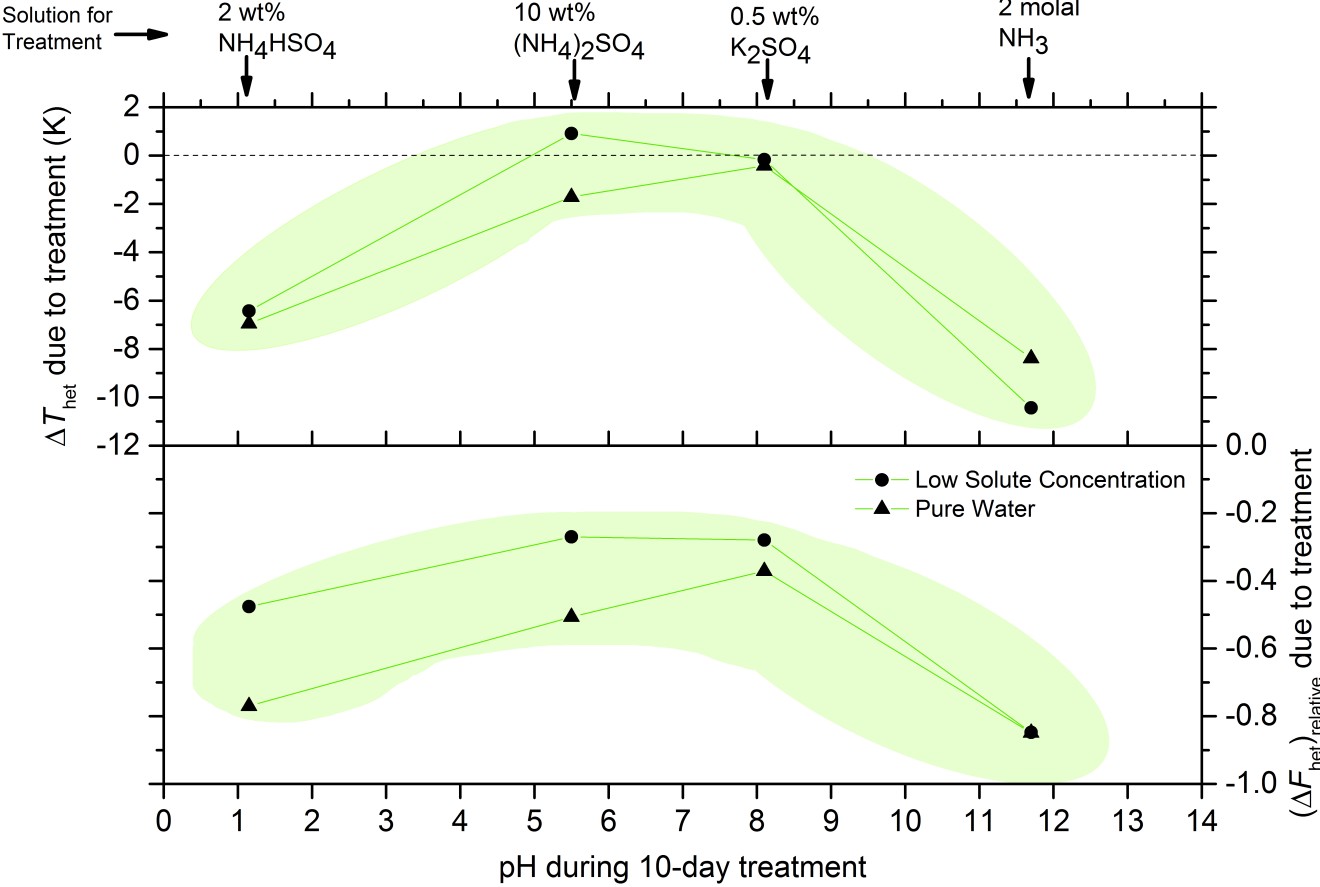

Figure 8. Aging of 2 wt% microcline suspended in 10 wt% $(NH_4)_2SO_4$, 2 wt% $NH_4HSO_4$, 0.5 wt% $K_2SO_4$, or 2 molal $NH_3$ solutions for 10 days, sorted according to the solution pH. After the treatment the particles were centrifuged, washed and resuspended in pure water or dilute solutions (0.1 wt% $(NH_4)_2SO_4$, 0.5 wt% $NH_4HSO_4$, 0.05 wt% $K_2SO_4$, 0.05 molal $NH_3$) and then freezing measured using DSC. Upper panel: difference in $T_{het}$ between particles treated in such a way and particles freshly suspended in pure water or very dilute solutions. Dashed black line: $\Delta T_{het} = 0$ shown to guide the eye. Lower panel: relative decrease in heterogeneously frozen fraction, $(\Delta F_{het})_{relative} = (F_{het,treated} - F_{het,fresh})/F_{het,fresh}$. The green shaded area in both panels depicts a suggested range for $\Delta T_{het}$ and $F_{het}$ based on the results from this study.

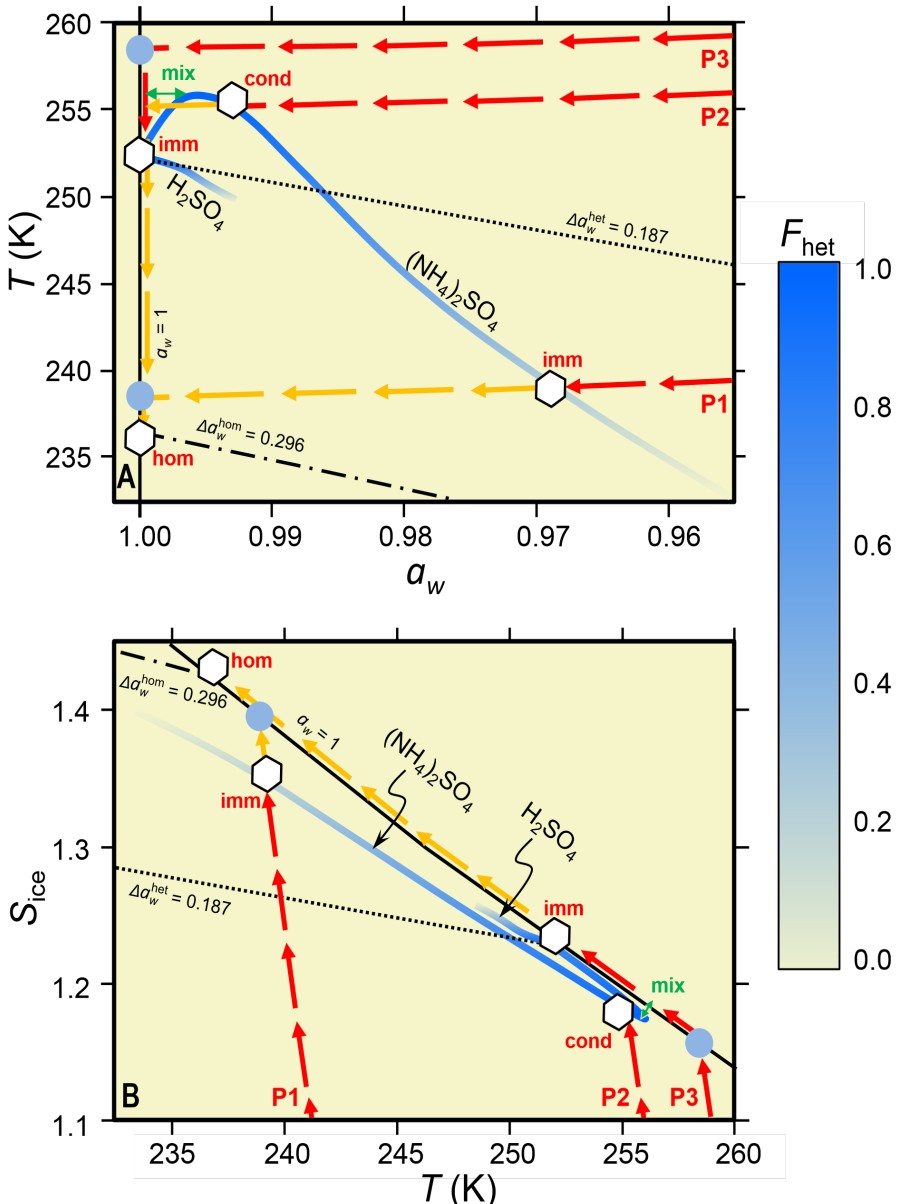

Figure 9. Phase diagrams, $a_w$ - $T$ (panel A) and $T$ - $S_{ice}$ (panel B), showing the development of aerosols containing microcline in dry (P1), moist (P2) and wet (P3) air parcels undergoing adiabatic cooling (red arrows) until ice nucleates (white hexagons). Blue curves: heterogeneous freezing onset temperatures, $T_{het}$, of 2 wt% microcline suspended in aqueous $(NH_4)_2SO_4$ or $H_2SO_4$ solutions. Color coding: heterogeneously frozen fraction, $F_{het}$, ranging from high (blue) to vanishing (yellow). In aqueous $(NH_4)_2SO_4$, ice can nucleate heterogeneously on microcline. In aqueous $H_2SO_4$, microcline is largely deactivated and ice nucleates homogeneously only after further cooling (yellow arrows) and forming a supercooled liquid water cloud (filled blue circles). Solid black line: water saturation ($a_w = 1$, RH = 100%). Dash-dotted black line: homogeneous ice nucleation, horizontally shifted from the ice melting curve by a constant offset $\Delta a_w^{hom} = 0.296$. Dotted black line: shifted by $\Delta a_w^{het} = 0.187$ derived from microcline suspended in pure water. Atmospheric freezing modes: "imm" = immersion; "cond" = condensation; "hom" = homogeneous. Entrainment/detrainment at cloud edges may cause nucleation at up to 4 K higher temperature than "imm" (green double arrow, "mix").

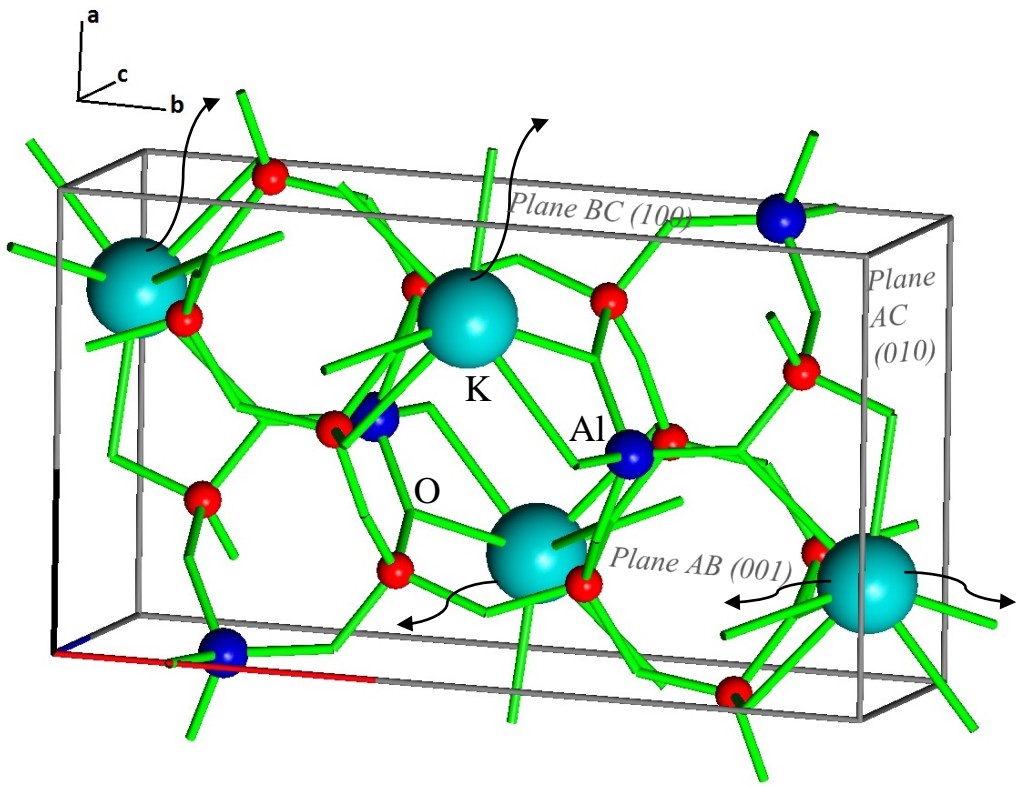

**125**

Figure A1. Simplified parallelepiped version of a primitive unit cell of microcline (chemical composition $K_4Al_4Si_{12}O_{32}$). Red spheres: Si atoms; blue spheres: Al atoms; cyan spheres: K atoms; green lines: -O- links between 2 neighboring Si/Al atoms. Black curved arrows show potential removal paths of $K^+$ from different planes of the unit cell (namely, AB, BC and AC). We assume that all $K^+$ from half of the unit cell are available for exchange, hence only the $K^+$ nearest to a plane can potentially move
**130**    out from that particular plane.

**135**

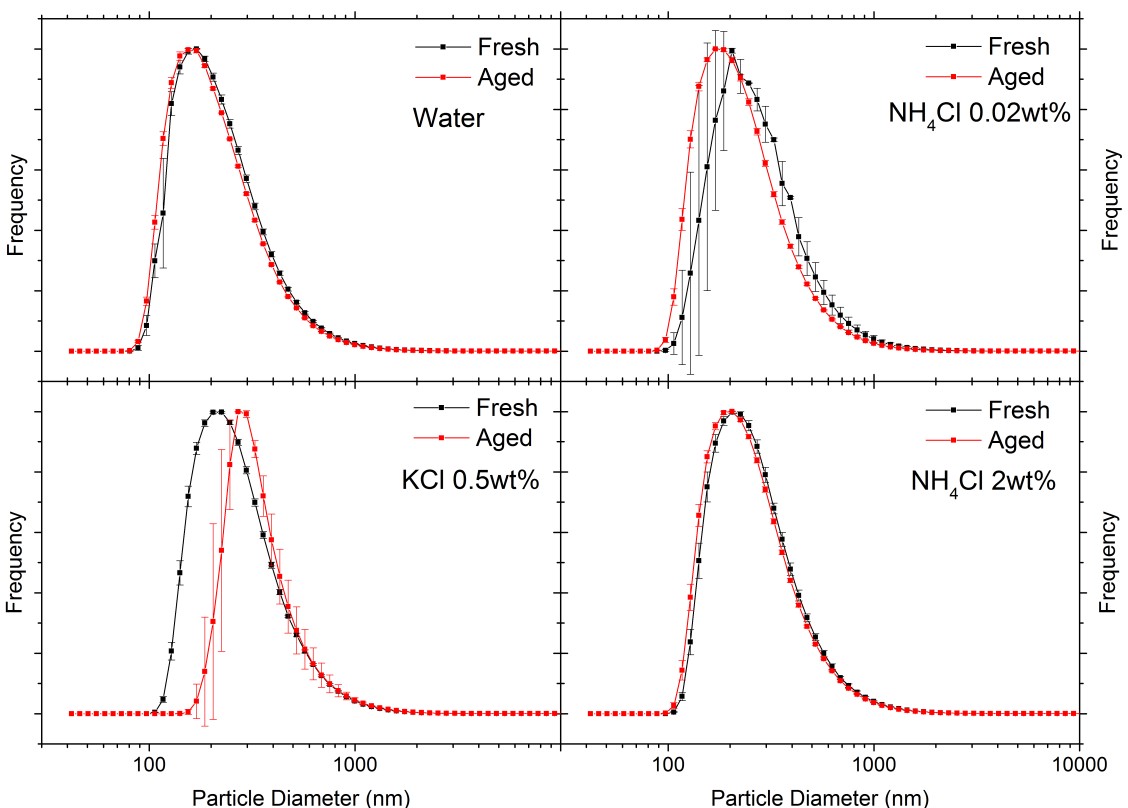

Figure B1. Particle size distributions obtained from the laser diffraction particle size analyser (Beckman Coulter LS13 320) for fresh (black squares connected with line) and aged suspensions (red squares connected with line) of 2 wt% microcline in pure water, 0.5 wt% KCl, 0.02 wt% $NH_4Cl$ and 2 wt% $NH_4Cl$. Size distribution measurements of each suspension were done in triplicates and each solid square shows the mean and one standard deviation as error bars. The aging period was 2 hours (roughly typical emulsion freezing measurement time span with DSC).