# Peer review of "Enhanced ice nucleation efficiency of microcline immersed in dilute NH3 and NH4+-containing solutions"

_Atmospheric Chemistry and Physics, 2018_

## Referee Comment (RC1) · Anonymous Referee #1 · 21 Feb 2018

**Review of "Enhanced ice nucleation efficiency of microcline immersed in dilute NH$_3$ and NH$_4^+$-containing solutions" by Kumar et al.**

**General Comment:**

This manuscript reports the ice nucleating abilities of K-feldspar microcline particles in the immersion freezing mode with the help of a Differential Scanning Calorimeter (DSC). With the goal to improve the current understanding of the good ice nucleating abilities of feldspar particles, K-feldspar microcline suspensions with different inorganic solutes such as NH$_3$, (NH$_4$)$_2$SO$_4$, NH$_4$HSO$_4$, NH$_4$NO$_3$, NH$_4$Cl, Na$_2$SO$_4$, H$_2$SO$_4$, K$_2$SO$_4$ and KCl were prepared and studied. Besides the ice nucleating abilities of microcline particles, the authors were also able to test the water activity approach, the effect of solute concentration, the role of adsorbed NH$_3$, the influence of aging, and the reversibility of surface modifications on their ice nucleating abilities. The authors found i) the heterogeneous freezing onset temperatures deviate from the previously established water-activity-based approach, ii) The good ice nucleating abilities of microcline particles cannot be attributed to neither the native surface K+ ions nor their exchange with externally added cations, iii) Chemically adsorbed ammonia molecules on the particle surface seems to play an important role in the ice nucleating abilities of microcline, iv) Aging could play an important role in the ice nucleating abilities of microcline particles depending on the solute, pH, and, solute concentration, and v) There is a possibility for condensation freezing to occur at a warmer temperatures than immersion freezing in dilute ammonia containing microcline droplets.

This is a well written manuscript, with very well designed and carefully conducted experiments. The current study provides very important evidences of the ice nucleating abilities of feldspar and brings our understanding one step forward. Given the importance of feldspar particles in ice cloud formation on a regional and global scale, this is a great contribution to the ice nucleation community. The reviewer did not identify any major point on the manuscript. This can be basically accepted as is; however, below is a short list of minor comments that can be considered for the final version.

**Minor comments:**

Line 39: Remove "the" after "precipitation in".

Line 40: Replace "remaining" with "existing".

Line 42: "nucleation mechanisms at work" sounds a bit awkward.

Line 44: "supercooled" or "supersaturated"?

Line 52: Replace "at work" with "to take place".

Line 72: Add a reference after "efficiency".

Line 73: Define "cloud glaciation". Why is this relevant for the manuscript?

Line 89: Replace "differential scanning calorimetry" with "DSC".

Line 95: Replace "Nannochloris atomus and Thalassiosira pseudonana" with "*Nannochloris atomus* and *Thalassiosira pseudonana*".

Line 96: Remove "dust" after "illite".

Line 108: Replace "differential scanning calorimetry" with "DSC".

Lines 125-126: "Evaluation" of what?

Lines 171-172: "The first signal observed at higher temperature is due to heterogeneous freezing triggered by microcline particles while the second freezing signal at lower temperature is due to homogeneous freezing." This was already mentioned above.

Lines 376-377: "is to large extent reversible" sounds a bit awkward.

Line 401: "minor contributions" on what?

Line 403: Replace "superior" with something more appropriate.

Line 424: "atmospheric solution droplets by means of Fig. 9" sounds a bit awkward.

Line 438: Remove "temperature" after "lower".

Line 444: Remove "the" before "nucleation".

Line 464: Replace "ice nucleation" with "IN".

Line 471: Replace "creation" with something more appropriate.

Line 483: Replace "ice nucleation" with "IN".

Line 624: Correct the volume and page number.

Line 822: Use the short name of the journal for consistency.

Figure 5: The readability of this figure is very low. Could the authors add and insert with a zoom of the figure down to aw of .99 or .98 to better visualize the data? Or can the authors present the data of this figure in a Table in the Appendix?

---

## Referee Comment (RC2) · Anonymous Referee #2 · 25 Feb 2018

General comments: I support publication of this manuscript in ACP. The observation of solutes' effects on the IN efficiency of microcline is novel and highly relevant to ACP as well as IN research community. The experimental approach employed in this study is elegant, and the authors conducted very careful and dedicated experimental works. Further elaborating/tightening the conclusion section with their observed results and detailed evidence would benefit the overall quality of the manuscript. Other than that, I have some minor comments that should be adequately addressed prior to the final publication.

Minor specific and technical comments: P2 L38: DeMott 2013a (a needs to come first)

[Figure]

P2 L67-68: This statement seems misleading - Harrison et al. (2016) points out that the fresh sanidine can be as active as microcline.

P2 L70-72: I suggest the authors to provide some additional information regarding the active site "assumption" (there is no direct observation of it in immersion freezing) and prescribe how it may be responsible for heterogeneous freezing in more detail here. Such information may benefit some statements in this manuscript (e.g., P9 L296-299; P14 L489) to sound less ambiguous.

P3 L74: "...included in the current IN parameterizations."?

P4 L115: The authors may want to introduce more of theoretical description of the water activity based immersion freezing approach (see Knopf et al., 2018 and references therein; doi: 10.1021/acsearthspacechem.7b00120) here first for the readers, who are not familiar with it.

P4 L130-132: The authors may briefly discuss the reproductively of onset temperatures, such as Thet, Thom and Tmelt, by DSC for the samples used in this study. This additional information may be beneficial to the readers, who are not familiar with the DSC technique.

P4 L144-P5 L150: Please clarify if the samples were re-sonicated or re-homogenized prior to each time trial DSC analysis. The presence/absence of any additional physical sample modifications should be stated.

P5 L164-167: How does this BET specific surface value compare to the other values from previous microcline IN studies? Did the authors observe any deviation between BET and the geometric surface-to-mass ratio (i.e., the ratio of the total surface area concentration to the total mass concentration estimated by SMPS/APS)? If so, what is its atmospheric implication?

P5 L172-174: I like this statement showing negligible concertation dependency of the onset freezing T in DSC. Great job.

P6 L205: Clarify/elaborate what "an initial increase" meant here – e.g., an increase in Thet near aw ∼1.0.

P8 L269-271 or P9 L320-321: The authors may extend this simulation-based discussion of ice-like ordering on the second bilayer, which seems to hold true for even non-mineral dust particles, by citing some more papers, such as Lupi et al. (2014, J. Am. Chem. Soc.) and Lupi and Molinero (2014, J. Phys. Chem. A).

P9 L314: "... when compared with ...".

P10 L328-332: I suggest the authors to separate this sentence into at least two sentences – e.g., "...K+ in the surface water. For instance, the comparison of...".

P10 L334-337: I suggest the authors to separate this sentence into two sentences – e.g., "...small structural fragments. This cations release presumably leads to... (Add proper references for this part if they are any)".

P10 L353-359: Please include the discussion of the recent study done by Abdelmonem et al. (2017, ACP, 7827-7837) regarding the effect of pH on IN.

P11 L369-372: This observation regarding non-reversible IN ability is great. Good job.

P12 L411: "An exception includes...".

Fig. 2: What does the minor heat flow peaks at T > 255 K indicate? It seems not apparent in Fig. 4. What caused the observed difference?

Fig. 3: Is there anyway to overlay the results of previous ABIFM results for some reference mineral dusts in comparison to the authors' results? If yes, please do. Adding some reference points may benefit the paper.
* * *

---

## Referee Comment (RC3) · Anonymous Referee #3 · 8 Mar 2018

Review for "Enhanced ice nucleation efficiency of microcline immersed in dilute NH3 and NH4+-containing solutions" by Kumar et al., submitted to ACPD

The manuscript examines the ice nucleation ability of microcline, a feldspar mineral, and how this changes when microcline particles are immersed in different solutions. This is done by using a DSC (differential scanning calorimeter). It is a thorough study describing fundamental processes and gaining interesting results.

However, I have a concern when it comes to the methodology and how results can be directly related to the atmosphere. Comparing different data sets obtained with the herein used DSC with each other is fine, as long as the droplet size distributions in

the different emulsions are the same. (The latter should be discussed more, and I elaborate on that below.) But details of the methodology influence the extent to which the obtained data can be used to derive atmospheric implications. Respective matters should be discussed in the text, which is mentioned below in more detail.

After these issues, together with some others listed below, will have been addressed, I can recommend the manuscript for publication in ACP.
* * *
Referring to the methodology and the relation to the atmosphere:

Some more information on the droplet size distributions in the emulsions would be good. The fact that not all droplets in the emulsions have the same size, and particularly that there likely are some large droplets with a high amount of material in them, will influence the extent to which the obtained data can be used to derive atmospheric implications.

In general, it is known that the ice nucleation ability of microcline particles depends on the minerals' surface area per droplet (Peckhaus et al., 2016, Harrison et al., 2016, Niedermeier et al., 2015). And the broadness of the peaks you obtain with DSC (e.g., Fig. 2) clearly shows that there is a broad distribution (of surface area per droplet) in your emulsions.

Broad droplet size distributions with few large droplets might explain why the onset temperature of heterogeneous freezing does not change much with microcline concentration (line 172-174): in all emulsions, there may have been a few large droplets with a comparably high microcline content (more precise: a high total microcline surface area per droplet), which were responsible for the onset of freezing. Interestingly, this temperature you report ($\sim$ 251 K) also is the temperature at which the strong increase in the freezing spectrum for microcline ("K-feldspar") in the paper by Atkinson et al. (2013) starts. The droplets with the highest content of microcline in your experiments likely are similar (in microcline surface area per droplet) to those examined by Atkinson et al. (2013).

I do take from your text that you yourself assume that there are multiple particles contained in single droplets, as you mention possible aggregation (line 288, lines 550-551). This is, however, only mentioned at these two occurrences, but might influence your results more broadly. This should be incorporated more whereever it could influence your results.

Has the DSC method been compared to single particle or freezing array methods before? If yes, this could simply be mentioned in the text, together with the results on how the different methods compare. In general, it would be good to know how large the droplets you looked at were on average, and how broad was their size distribution? And how broad was the particle size distribution? And how were the particles distributed to the droplets? Or, summarized in one parameter: how was the mineral surface area distributed to the droplets? Is anything known on that? If yes, please add this. If not, please at least mention this and discuss the implications. One implication is, that you cannot directly transfer your results to the atmosphere, where each droplet will always contain a single (comparably small) particle.

And last but not least: How reproducible are the distributions in the emulsions? And how reliably can the freezing spectra be evaluated (as e.g., those shown in Fig. 4)? And what is the uncertainty of the derived values?

Related to that are also the following two remarks:

line 345: You observe that F_het decreases during the first day after microcline was suspended in pure water, while T_het was preserved. Could a reason be that droplets are settling out? Again, it seems that the majority of your droplets might act different than the few ones that determine the freezing onset. And in this respect, if the different emulsions had different surface area distributions for the experiments in pure water and in dilute NH3 and (NH4)2SO4 solutions, this may also explain observed differences.

Please discuss this shortly, too.

line 372: Here, too, different surface area distributions might influence F_het, similar to the point mentioned directly above.

As a bottom line of all of that said above (and besides for revisions in the text mentioned above), the direct translation of your results to the atmosphere, even with giving degrees of Kelvin by which the occurrence of freezing may be shifted, needs to be discussed more critically or may even be shortened. Your results on the influence on the surfaces and surface sites alone is already a valuable contribution.
* * *
General comments:

line 51ff: Below, you discuss that deposition ice nucleation was questioned by Marcolli (2014), and similarly, for condensation freezing, you should also include that Vali et al. (2015) says: "Whether condensation freezing on a microscopic scale, if it occurs, is truly different from deposition nucleation, or distinct from immersion freezing, is not fully established." This is also related to line 442, where you use "condensation freezing", which, however, following the definition by Vali et al. (2015) given in your introduction does certainly not take place in your DSC measurements. But you used these measurements to make up your scenarios. Please be consistent!

line 276-278: Is it really so improbable that the (100) surface is exposed? - After all, you milled your samples, and the number density of active sites in microcline is "only" $\sim 1000000$ cm-3 at 251 K in Atkinson et al. (2013). This site density would need to be compared to the expected number of cracks and defaults that may occur during milling, before this statement you make here can be made. BTW: this is again a point where it would be good to know the exact distributions of droplet sizes and of particles in the droplets.

The whole chapter 4.5: This chapter made a somewhat unfinished impression on me.

How do the results here fit in line with what you described earlier? And as you only did the experiments at a single (and always different) pH for each of the solutions, a difference between the effect of the dissolved substance versus the pH cannot be obtained. This should also be mentioned in the text (e.g., connected to what you write in line 381). Also, in the end of Chapter 4.5, you cite a number of studies, however, without putting them in context to your results, so while reading this part of your text, I got confused. Similarly, when later reading the part on aging in the conclusions (line 501 ff) I was astounded as this did not reflect what I took from this chapter. Please revise these parts of the text.

line 410: You state that "Saharan dust particles undergo little chemical processing during long-range transport across the Atlantic unless they become incorporated in cloud droplets". However, dust particles are CCN in the atmosphere (Karydis et al., 2011), so I wonder if you want to say that dust particles do not act as CCN, or that they do not become incorporated in cloud droplets because Saharan air masses are so dry that clouds do not form? Please clarify, and make clear that dust particles are CCN.

line 435: (Again:) Your method prohibits to make statements about single particles - and, strictly speaking, also about atmospheric onset temperatures, as a single (and then likely smaller) particle in the atmosphere will only activate ice at lower temperature. I know that I mentioned this before. But again, I urge you to state this clearly.

line 505: Let me ask you a question: Have you ever observed ice crystals outside of clouds in the mixed phase cloud temperature range (unless they fall out from a cloud)? What you suggest here might suggest that this could be observed. Based on the fact that your method rather is a bulk method and not one for single particles, I (again) suggest you are careful in drawing conclusions for processes going on in the atmosphere.
* * *
Technical corrections:

line 45: The abbreviation "IN" is used but has only been defined in the abstract. I'd define it again on the first appearance in the text.

line 72: You say "the particle", but in this context, it is not clear, which particle you mean.

line 115: Replace "the" with "a", as this is where the setup is first introduced. Also, DSC was defined in abstract, but I'd define it again in main text upon its first appearance, ideally together with a citation where it is described in detail.

line 115: Upon reading this the first time, I wished for more information on the microcline when it was first mentioned here, particularly as you give all the detailed information about all the chemicals here, too. Now I know that this is given in 2.4. – maybe you could swap the chapters, so that 2.4 comes first, or you could at least mention here that there is more on the microcline sample later.

line 120: Again, this is the first time that emulsions are mentioned, so delete "the".

line 155: Delete the "," following "Microcline".

line 179-180: There is something wrong with this sentence, please correct. Looks like a copy/paste error to me.

line 275: Kiselev et al. was published online 2016, and I have a version downloaded 2017 that says "cite as ..., 2016" - please check which year is correct.

line 298: I'd prefer "suggest" to "conclude". BTW: In line 296, you say that "excess solute strength hampers the IN efficiency". Do you have any idea, why that would be? If you could add a sentence on that here, I'd appreciate it.

line 310: Please add at or above which water activity you are referring to, here, in this sentence, as data all agree quite well in the lower concentration range.

Literature: Atkinson, J. D., B. J. Murray, M. T. Woodhouse, T. F. Whale, K. J. Baustian, K. S. Carslaw, S. Dobbie, D. O'Sullivan, and T. L. Malkin (2013), The importance of

feldspar for ice nucleation by mineral dust in mixed-phase clouds, Nature, 498(7454), 355-358, doi:10.1038/nature12278.

Harrison, A. D., T. F. Whale, M. A. Carpenter, M. A. Holden, L. Neve, D. O'Sullivan, J. V. Temprado, and B. J. Murray (2016), Not all feldspars are equal: a survey of ice nucleating properties across the feldspar group of minerals, Atmos. Chem. Phys., 16, 10927–10940, doi:10.5194/acp-16-10927-2016.

Karydis, V. A., P. Kumar, D. Barahona, I. N. Sokolik, and A. Nenes (2011), On the effect of dust particles on global cloud condensation nuclei and cloud droplet number, J. Geophys. Res.-Atmos., 116(D23204), doi:10.1029/2011jd016283.

Kiselev, A., F. Bachmann, P. Pedevilla, S. J. Cox, A. Michaelides, D. Gerthsen, and T. Leisner (2016), Active sites in heterogeneous ice nucleation—the example of K-rich feldspars, Science, doi:10.1126/science.aai8034.

Marcolli, C. (2014), Deposition nucleation viewed as homogeneous or immersion freezing in pores and cavities, Atmos. Chem. Phys., 14(4), 2071-2104, doi:10.5194/acp-14-2071-2014.

Niedermeier, D., S. Augustin-Bauditz, S. Hartmann, H. Wex, K. Ignatius, and F. Stratmann (2015), Can we define an asymptotic value for the ice active surface site density for heterogeneous ice nucleation?, J. Geophys. Res., 120, doi:10.1002/2014JD022814.

Peckhaus, A., A. Kiselev, T. Hiron, M. Ebert, and T. Leisner (2016), A comparative study of K-rich and Na/Ca-rich feldspar ice-nucleating particles in a nanoliter droplet freezing assay, Atmos. Chem. Phys., 16, 11477-11496, doi:10.5194/acp-16-11477-2016.

Vali, G., P. J. DeMott, O. Moehler, and T. F. Whale (2015), Technical Note: A proposal for ice nucleation terminology, Atmos. Chem. Phys., 15(18), 10263-10270, doi:10.5194/acp-15-10263-2015.

---

## Author Comment (AC1) · 19 Apr 2018

We thank Reviewer 1 for his/her constructive comments. We reproduce reviewer's comments in *blue* and our responses in black.

*Line 39: Remove "the" after "precipitation in".*

Suggested change has been made (line 40).

*Line 40: Replace "remaining" with "existing".*

Suggested change has been made (line 40).

*Line 42: "nucleation mechanisms at work" sounds a bit awkward.*

The term "nucleation mechanisms at work" has been changed to "nucleation mechanisms" (line 42).

*Line 44: "supercooled" or "supersaturated"?*

We replaced "supercooled" by "metastable" (line 44).

*Line 52: Replace "at work" with "to take place".*

Suggested change has been made (line 54).

*Line 72: Add a reference after "efficiency".*

"Kaufmann, L., Marcolli, C., Hofer, J., Pinti, V., Hoyle, C. R., and Peter, T.: Ice nucleation efficiency of natural dust samples in the immersion mode, Atmos. Chem. Phys., 16, 11177-11206, doi:10.5194/acp-16-11177-2016, 2016" has been added as a reference after "efficiency".

*Line 73: Define "cloud glaciation". Why is this relevant for the manuscript?*

We mean with "cloud glaciation" the transformation of cloud droplets to ice crystals. To understand cloud glaciation, the role of atmospheric processing on the IN efficiency of INPs such as microcline needs to be understood.

*Line 89: Replace "differential scanning calorimetry" with "DSC".*

This is the first mentioning of DSC in the main text, therefore, we want to state the full expression here. We add DSC in brackets in the revised manuscript (line 96).

*Line 95: Replace "Nannochloris atomus and Thalassiosira pseudonana" with "Nannochloris atomus and Thalassiosira pseudonana".*

Suggested change has been made (line 102).

*Line 96: Remove "dust" after "illite".*

Suggested change has been made (line 103).

*Line 108: Replace "differential scanning calorimetry" with "DSC".*

Suggested change has been made (line 117).

*Lines 125-126: "Evaluation" of what?*

Lines 125-126 of the discussion paper have been modified to "We ran the second freezing cycle at 1 K min$^{-1}$ cooling rate and used it for evaluation of freezing temperature ($T_{het}$ and $T_{hom}$) and heterogeneously frozen fraction ($F_{het}$) as discussed below." (lines 152-154). The evaluation of these parameters are discussed in the next paragraph.

*Lines 171-172: "The first signal observed at higher temperature is due to heterogeneous freezing triggered by microcline particles while the second freezing signal at lower temperature is due to homogeneous freezing." This was already mentioned above.*

This sentence has been removed from the revised manuscript.

*Lines 376-377: "is to large extent reversible" sounds a bit awkward.*

The part "is to large extent reversible" has been modified to "largely reversible" (line 434).

*Line 401: "minor contributions" on what?*

To improve the formulation, we write in the revised manuscript "the minor contributions of feldspars to atmospheric dust aerosols" (line 449).

*Line 403: Replace "superior" with something more appropriate.*

The term "superior" is replaced by "high" in the revised manuscript.

*Line 424: "atmospheric solution droplets by means of Fig. 9" sounds a bit awkward.*

We reformulate in the revised manuscript: "atmospheric solution droplets following atmospheric air parcel trajectories P1 through P3 (red and yellow arrows) with increasing moisture (as shown in Fig. 9)." (lines 471-473)

*Line 438: Remove "temperature" after "lower".*

Suggested change has been made.

*Line 444: Remove "the" before "nucleation".*

Suggested change has been made.

*Line 464: Replace "ice nucleation" with "IN".*

Suggested change has been made.

*Line 471: Replace "creation" with something more appropriate.*

The term "creation" has been replaced by "emergence" in the revised manuscript (line 520).

*Line 483: Replace "ice nucleation" with "IN".*

Suggested change has been made.

*Line 624: Correct the volume and page number.*

Necessary corrections have been made in the revised manuscript. "n/a-n/a" has been removed from the reference (line 684).

*Line 822: Use the short name of the journal for consistency.*

Journal name "Environmental Earth Sciences" has been abbreviated to "Environ. Earth Sci." in the revised manuscript (line 898-899).

*Figure 5: The readability of this figure is very low. Could the authors add and insert with a zoom of the figure down to aw of .99 or .98 to better visualize the data? Or can the authors present the data of this figure in a Table in the Appendix?*

All data for Figure 5 (as well as Figs. 3 and 7) are presented in a tabular form at the following DOI: *10.3929/ethz-b-000229892* and has been mentioned in "*Section 7. Data availability*".

---

## Author Comment (AC2) · 19 Apr 2018

We thank Reviewer 2 for his/her constructive comments. We reproduce the reviewer's comments in *blue* and our responses in black.

*P2 L38: DeMott 2013a (a needs to come first)*

We introduced DeMott 2013a at this position to follow the order. We added also Stephens et al. (1990) (Stephens, G. L., Tsay, S.-C., Jr., P. W. S., and Flatau, P. J.: The relevance of the microphysical and radiative properties of cirrus clouds to climate and climatic feedback, J. Atmos. Sci., 47, 1742-1754, doi:10.1175/1520-0469(1990)047<1742:trotma>2.0.co;2, 1990). (line 38)

*P2 L67-68: This statement seems misleading - Harrison et al. (2016) points out that the fresh sanidine can be as active as microcline.*

We have removed the reference "Harrison et al. (2016)" from the concerned statement. The statement "Microcline has been reported to be IN active at higher temperatures than orthoclase and sanidine" is supported by Kaufmann et al. (2016). (lines 70-71)

*P2 L70-72: I suggest the authors to provide some additional information regarding the active site "assumption" (there is no direct observation of it in immersion freezing) and prescribe how it may be responsible for heterogeneous freezing in more detail here. Such information may benefit some statements in this manuscript (e.g., P9 L296-299; P14 L489) to sound less ambiguous.*

Active sites, nucleation sites, or, just sites have become a very common concept in the discussion of ice nucleation in the atmosphere. In Vali et al. (2015), a site is defined as a "preferred location for ice nucleation on an INP, or equivalent". Although sites responsible for immersion freezing cannot be directly observed, their existence is plausible considering refreeze experiments, which show statistically relevant differences between the freezing temperatures of samples taken from the same stock suspension when they are subjected to repeated freezing cycles (Vali, 2014; Kaufmann et al., 2017). Such differences cannot be explained assuming that a uniform surface common to all particles of a sample is responsible for freezing. An active site needs to be large enough to accommodate a critical ice embryo. Applying classical nucleation theory, Kaufmann et al. (2017) determined active sites with areas of $15 - 50$ nm$^2$ to be responsible for ice nucleation on mineral dusts. In the revised manuscript, we explain the concept of active sites in the introduction starting on line 49: "Statistical analysis of freezing experiments indicates that heterogeneous ice nucleation is induced by active sites present on a foreign body termed as ice-nucleating particle (INP). Active sites are preferred

locations for ice nucleation with minimum areas in the order of $10 - 50$ nm$^2$ based on estimates using classical nucleation theory (Vali, 2014; Vali et al., 2015; Kaufmann et al., 2017). (lines 49-52)

*P3 L74: "...included in the current IN parameterizations."?*

We improve the formulation by writing: "needs to be taken into account".

*P4 L115: The authors may want to introduce more of theoretical description of the water activity based immersion freezing approach (see Knopf et al., 2018 and references therein; doi: 10.1021/acsearthspacechem.7b00120) here first for the readers, who are not familiar with it.*

More discussion on *water activity based immersion freezing approach* has been added in the "Introduction". Lines 86-89: "Describing the kinetic (non-equilibrium) IN process as a simple function of a thermodynamic (equilibrium) quantity $\Delta a_w$ is tempting because of its simplicity. Therefore, the applicability of "*water-activity criterion*" proposed by Koop et al. (2000) for homogeneous IN to heterogeneous IN temperatures of various types of INPs in immersion mode has been probed by several studies in the past." And lines 105-107: "Several other studies that have applied this approach to describe heterogeneous IN include Zobrist et al. (2006), Knopf et al. (2011) and Archuleta et al. (2005) (see Knopf et al. (2018) for a detailed review on this topic)."

*P4 L130-132: The authors may briefly discuss the reproductively of onset temperatures, such as Thet, Thom and Tmelt, by DSC for the samples used in this study. This additional information may be beneficial to the readers, who are not familiar with the DSC technique.*

We experimentally determine the reproducibility be repeating all experiments at least once. We give the experimental variability that includes emulsion preparation and DSC measurements in the figure captions by stating e.g. for Fig. 3: "Symbols are the mean of at least two separate emulsion freezing experiments. Three symbols carry error bars to show representative experimental variations (min-to-max) in $T_{het}$ and $a_w$."

We add this information also to the methodology section 2.2 of the revised manuscript: "Average precisions in $T_{het}$ are $\pm0.15$ K with maximum deviations not exceeding 0.8 K. $T_{hom}$, and $T_{melt}$ are precise within $\pm0.1$ K. Absolute uncertainties in $F_{het}$, are on average $\pm\ 0.02$ and do not exceed $\pm$ 0.1." (lines 165-167)

*P4 L144-P5 L150: Please clarify if the samples were re-sonicated or re-homogenized prior to each time trial DSC analysis. The presence/absence of any additional physical sample modifications should be stated.*

The suspensions were sonicated each time before preparing emulsion. This information is now

mentioned in the revised manuscript on lines 172-173: "Each suspension was sonicated for 5 min before preparing the emulsions in order to re-suspend the particles that have settled over time and avoid particle aggregation."

*P5 L164-167: How does this BET specific surface value compare to the other values from previous microcline IN studies? Did the authors observe any deviation between BET and the geometric surface-to-mass ratio (i.e., the ratio of the total surface area concentration to the total mass concentration estimated by SMPS/APS)? If so, what is its atmospheric implication?*

Atkinson et al., (2013) report a BET surface area of 3.2 m$^2$/g for their K-feldspar, a microcline, while the BET area of our microcline is 1.9 m$^2$/g. These surface areas are similar to the ones of quartz (0.5 – 5.5 m$^2$/g, Zolles et al., 2015), but lower than surface areas of kaolinites (10 - 25 m$^2$/g, Pinti et al., 2012), and much lower than surface areas of illites (~100 m$^2$/g, Pinti et al., 2012) and montmorillonites (30 – 84 m$^2$/g, Pinti et al., 2012).

The specific surface area values evaluated from the laser diffraction particle analyser (LDPS) and SMPS/APS is 1.21 m$^2$/g and 2.43 m$^2$/g, respectively. From these minor differences we don't expect atmospheric implications.

*P5 L172-174: I like this statement showing negligible concentration dependency of the onset freezing T in DSC. Great job.*

We thank the reviewer for the appreciation.

*P6 L205: Clarify/elaborate what "an initial increase" meant here – e.g., an increase in Thet near aw~1.0.*

The intention was to highlight the increase in $T_{het}$ near $a_w$ ~1. The sentence has been modified to: "For all $NH_4^+$ solute cases in dilute concentrations viz. $NH_3$ (< 1 molal), $(NH_4)_2SO_4$ (< 0.16 molal), $NH_4HSO_4$ (< 0.18 molal), $NH_4NO_3$ (< 0.67 molal) and $NH_4Cl$ (< 0.99 molal), there is an increase in $T_{het}$ compared to $T_{het}^{\Delta a_w^{het}}(a_w)$ near $a_w$ ~ 1." (lines 217-219)

*P8 L269-271 or P9 L320-321: The authors may extend this simulation-based discussion of ice-like ordering on the second bilayer, which seems to hold true for even nonmineral dust particles, by citing some more papers, such as Lupi et al. (2014, J. Am. Chem. Soc.) and Lupi and Molinero (2014, J. Phys. Chem. A).*

We thank the reviewer for the suggestion. The references have been added and discussed in the revised manuscript (lines 320-321) as "Using molecular dynamics simulations, Lupi et al. (2014)

and Lupi and Molinero (2014) have recently shown that even non-mineral graphitic surfaces can promote IN due to surface-induced ordered layering of interfacial water."

*P9 L314: ". . . when compared with . . .".*

Suggested change has been made.

*P10 L328-332: I suggest the authors to separate this sentence into at least two sentences – e.g., ". . .K+ in the surface water. For instance, the comparison of. . .".*

As suggested by the reviewer, the lines (328-332) have now been split into 3 shorter sentences as "The total absence of IN activity at higher $K^+$ concentrations indicates that the replacement of $K^+$ with $H^+/H_3O^+$ is essential for the IN activity of microcline. This suggests that the higher IN efficiency of K-feldspars compared with (Na, Ca)-feldspars does not stem from the beneficial effect of $K^+$ in the surface layer. For instance, the comparison of $K_2SO_4$ with $Na_2SO_4$ solutions with the same water activity shows that the presence of $K^+$ has even a more negative effect on $F_{het}$ than $Na^+$." (lines 377-382)

*P10 L334-337: I suggest the authors to separate this sentence into two sentences – e.g., ". . .small structural fragments. This cations release presumably leads to. . . (Add proper references for this part if they are any)".*

The suggested lines have been modified and split into 2 separate sentences as "Suspension of feldspars in water leads to the immediate release of surface cations within minutes (Nash and Marshall, 1957; Busenberg and Clemency, 1976; Smith, 1994; Peckhaus et al., 2016). Over a period of years, the continuous release of silica as silicic acid, hydrated alumina and small structural fragments leads to the slow disintegration of feldspar and to the buildup of new minerals (DeVore, 1957; Banfield and Eggleton, 1990)." (lines 384-387)

Two new references have been added to the second sentence, viz:

DeVore, G. W.: The surface chemistry of feldspars as an influence on their decomposition products, Clays Clay Miner., 6, 26-41, doi:10.1346/CCMN.1957.0060104, 1957.

Banfield, J. F., and Eggleton, R. A.: Analytical transmission electron microscope studies of plagioclase, muscovite, and k-feldspar weathering, Clays Clay Miner., 38, 77-89, doi:10.1346/CCMN.1990.0380111 1990.

*P10 L353-359: Please include the discussion of the recent study done by Abdelmonem et al. (2017, ACP, 7827-7837) regarding the effect of pH on IN.*

We refer to the study by Abdelmonem by stating: "The solution pH influences the surface charge of hydroxylated mineral surfaces. This again influences the ordering of water molecules at the water/mineral interface and the IN efficiency, as was recently shown by Abdelmonem et al. (2017). In case of feldspars, the pH of the solution is, in addition, an important parameter in determining the stability of the feldspar surface." (lines 403-405)

*P11 L369-372: This observation regarding non-reversible IN ability is great. Good job.*

We thank the reviewer for the appreciation.

*P12 L411: "An exception includes. . .".*

Suggested change has been made.

*Fig. 2: What does the minor heat flow peaks at T > 255 K indicate? It seems not apparent in Fig. 4. What caused the observed difference?*

The minor heat flow peaks/spikes at T > 255 K are heterogeneous freezing signals of particularly large single droplets in the tail of the droplet size distribution, which are not representative for the sample. Their presence is due to the coalescence of some smaller droplets probably while transferring the sample to the aluminum pan for DSC and is not reproducible. We mention them at the end of Secti. 2.2 of the revised manuscript. In addition, we add to the figure caption of Fig. 2: "Note that the spikes at $T > 255$ K are due to the freezing of single particularly large emulsion droplets. They are not reproducible and therefore excluded from evaluation."

*Fig. 3: Is there any way to overlay the results of previous ABIFM results for some reference mineral dusts in comparison to the authors' results? If yes, please do. Adding some reference points may benefit the paper.*

We plot for comparison the heterogeneous freezing curve under the assumption that heterogeneous freezing can be predicted by a constant shift of the melting curve. This corresponds with the assumption of the activity based immersion freezing model (ABIFM) proposed by Knopf and Alpert (2013). INP that follow this assumption lie along this line. Such curves can e.g. be seen in Fig. 5 of Zobrist et al. (2008) and in Fig. 2 of Knopf and Alpert (2013). We consider that the figure would become too busy and confusing if we included results from other INPs. We mention in the discussion on line 267 of the revised manuscript: "Note that this derivation of the heterogeneous freezing temperatures on the basis of the solution water activity and the freezing temperature of the suspension in pure water is analogous to the one of ABIFM and results in the same prediction." (lines 267-269)

**References**

[revised manuscript text omitted]

---

## Author Comment (AC3) · 19 Apr 2018

We thank Reviewer 3 for his/her constructive comments. We reproduce the reviewer's comments in *blue* and our responses in black in this document.

*Some more information on the droplet size distributions in the emulsions would be good. The fact that not all droplets in the emulsions have the same size, and particularly that there likely are some large droplets with a high amount of material in them, will influence the extent to which the obtained data can be used to derive atmospheric implications.*

This is a good point. The emulsion preparation procedure has been developed in our group and was first described in Marcolli et al. (2007). It has been used in Pinti et al. (2012) and Kaufmann et al. (2016). Droplet size distributions resulting from this procedure are shown in Figs. 1 of Marcolli et al. (2007), Pinti et al. (2012) and Kaufmann et al. (2016). Our size distribution are comparable with these. To make this clear, we add to Sect. 2.2 of the revised manuscript on lines 148-150 the following sentences:

"This procedure leads to droplet size distributions peaking at about 2 – 3 µm in number and a broad distribution in volume with highest values between 4 and 12 µm similar as the ones shown in Figs. 1 of Marcolli et al. (2007), Pinti et al. (2012), and Kaufmann et al. (2016)."

For 2 wt % microcline suspensions, the smallest droplets of emulsions prepared by this procedure remain empty while the largest ones carry in the order of 1000 microcline particles. We consider these largest droplets relevant for the onset freezing temperatures. Since these droplets freeze due to a nucleation event occurring on the best of about 1000 particles, the onset freezing temperature has a high bias compared with the freezing temperature of an average microcline particle. This may explain the higher freezing temperatures observed in our experiments compared with the ones from single particle freezing experiments (Niedermeier et al., 2015; Burkert-Kohn et al., 2017).

*Broad droplet size distributions with few large droplets might explain why the onset temperature of heterogeneous freezing does not change much with microcline concentration (line 172-174): in all emulsions, there may have been a few large droplets with a comparably high microcline content (more precise: a high total microcline surface area per droplet), which were responsible for the onset of freezing. Interestingly, this temperature you report (~ 251 K) also is the temperature at which the strong increase in the freezing spectrum for microcline ("K-feldspar") in the paper by Atkinson et al. (2013) starts. The droplets with the highest content of microcline in your experiments*

*likely are similar (in microcline surface area per droplet) to those examined by Atkinson et al. (2013).*

A constant onset temperature of heterogeneous freezing is indeed a sign of a narrow distribution of active site freezing temperatures. There are examples of a shift of freezing onset to warmer temperatures with increasing suspension concentration. Such an increase is observed for ATD, which is a mixture of minerals (see Fig. 4a of Marcolli et al. (2007)). When a minor component is freezing at warmer temperatures than the dominant species, a second heterogeneous freezing peak appears for higher suspension concentrations as is the case for kaolinite from Sigma Aldrich (same as Fluka) containing a minor component of feldspars. The mineralogical pure kaolinite from the Clay Mineral Society KGa-1b has an almost constant onset (see Fig. 3 of Pinti et al., 2012). Atkinson et al. (2013) report an active site density of about $10^6$ cm$^{-2}$ for microcline between 251 K and 252 K. The freezing onset for 2 wt% microcline suspensions is ~252 K (see Fig. 3.). If we assume that all the microcline particles in our study are of average size (mode diameter of 213 nm), droplets responsible for the freezing onset (12 µm diameter) should contain about 1000 particles in case of 2 wt% suspensions resulting in an active site density of at least $5 \cdot 10^{-5}$ cm$^{-2}$ at ~252 K. The same evaluation for 0.2 wt% and 20 wt% microcline emulsions yields active site densities of at least $5 \cdot 10^{-6}$ cm$^{-2}$ and at least $5 \cdot 10^{-4}$ cm$^{-2}$ at 251 K and 252 K, respectively. Note, that it is only possible to determine a lower limit of active site densities because more than one site may be responsible for freezing especially at high suspension concentrations. Overall, these numbers are in excellent agreement with Atkinson et al. (2013) considering the roughness of the estimate and that microcline samples from different sources and origins have been compared. Moreover, it shows that microcline exhibits a characteristic active site with little variation in the freezing temperature.

The above mentioned discussion has been added as a new section to the revised manuscript: **"Section 4.1 Heterogeneous ice nucleation of microcline in pure water"** (Lines 233-263)

*I do take from your text that you yourself assume that there are multiple particles contained in single droplets, as you mention possible aggregation (line 288, lines 550-551). This is, however, only mentioned at these two occurrences, but might influence your results more broadly. This should be incorporated more wherever it could influence your results.*

We rather think that aggregation occurs in the suspension before the emulsions are prepared and could lead to more empty droplets. We therefore sonicate the suspensions to avoid possible aggregation before the preparation of the emulsions. We have discussed this in Appendix B in the

manuscript where we determined the particle size distribution of microcline suspended in pure water with a laser diffraction particle size analyzer. Suspensions were sonicated for 2 minutes before the size distribution measurement. Negligible agglomeration was observed during aging for 2 hours (Figure B1). Therefore, we assume negligible aggregation also within the emulsion droplets.

*Has the DSC method been compared to single particle or freezing array methods before? If yes, this could simply be mentioned in the text, together with the results on how the different methods compare.*

We made such comparisons in Kaufmann et al. (2016). Through emulsion freezing experiments the whole distribution of particles is investigated for their IN efficiency without any extra information concerning the dependence on particle size. In Kaufmann et al. (2016), we applied the active site parameterization developed by Marcolli et al. (2007) based on DSC freezing experiments with polydisperse ATD (Arizona Test Dust) to other studies that investigated the IN efficiency of ATD (Niedermeier et al., 2010; Hoyle et al., 2011; Nagare et al., 2016). We found throughout consistency, e.g. a low ice active fraction for 300 nm ATD particles. This confirms our assumption of random distribution of INP within droplets that was the basis of the simulations in Marcolli et al. (2007). Niedermeier et al. (2015) observed onset freezing temperatures of 248 K in single particle immersion freezing experiments (LACIS) with microcline. Their onset corresponds to a frozen fraction of 0.02 to 0.05 and does not vary much over size-segregated K-feldspar (76% microcline) particles with diameters from 200 nm to 500 nm. The 500 nm particles achieve an ice active fraction equal to one, while the smaller particles reach a plateau at 238 K below an ice active fraction of one. Burkert-Kohn et al. (2017) performed single particle immersion freezing experiments with size-segregated microcline and found a frozen fraction of 0.1 up to 253 K for 300 nm particles and up to 248 K for 200 nm particles. Our microcline sample consists of particles with diameters ranging from 100 nm to 2 µm and a mode diameter determined as 213 nm and exhibits $T_{onset}$ of 251 – 252 K for suspensions in pure water with concentrations between 0.2 wt% and 20 wt%. Kaufmann et al. (2016) investigated two microcline samples with onset freezing temperature range of 250.5 – 251.8 K (microcline from Namibia) and 251.8 - 252.8 K (microcline from Elba) for suspension concentrations of 0.5 – 10 wt%. Kaufmann et al, (2016) calculated active particle fractions of 0.64 for Micrcline Elba and 0.54 for Microcline Namibia, but concluded that considering the uncertainties associated with these estimates, the data would also be consistent with an active particle fraction of one. In conclusion, the DSC results are consistent with results from single particle and droplet freezing setups, even more if

one considers the variability of IN activities between different microcline samples as shown in Fig. 5 by Harrison et al. (2016).

The new section (**Sect. 4.1**) in the revised manuscript discusses the comparison of our freezing onsets in pure water with the ones of other studies. (Lines 244-263)

*In general, it would be good to know how large the droplets you looked at were on average, and how broad was their size distribution? And how broad was the particle size distribution? And how were the particles distributed to the droplets? Or, summarized in one parameter: how was the mineral surface area distributed to the droplets? Is anything known on that?*

The size distribution of dry dispersed microcline particles evaluated using APS/SMPS has been added in the revised Supplementary Material (Figure S17). Moreover, the size distribution of microcline particles suspended in water is shown in Fig. B1, upper left panel.

The probability for a droplet to contain at least one particle depends on its volume. Therefore, the probability for a larger water droplet to freeze heterogeneously is higher than for a smaller water droplet. Hence, freezing of larger droplets dominates the heterogeneous freezing signal and freezing of smaller droplets the homogeneous freezing signal. We have discussed this in more detail previously in Kaufmann et al (2016) *(Sect. 3 Statistical evaluation of emulsion measurements)* as well as in the current manuscript *"Sect. 3.1 Effect of microcline concentration on the heterogeneous freezing signal"*. We extend this discussion in the revised manuscript (Lines 191-196):

"For 20 wt% microcline suspensions, droplets with diameters of 1.1µm are on average filled with one microcline particle, while smaller droplets are on average empty. This number shifts to 6.3 µm for 0.2 wt% microcline suspensions. 2 wt% suspensions were picked for further investigations leading to a good heterogeneous freezing signal with little particle agglomeration (see also Kaufmann et al. (2016) and Appendix B). Droplets with diameters of about 12 µm are considered to be relevant for the freezing onset in the DSC. For a 2 wt% suspension of microcline, these droplets contain about 1000 microcline particles."

*And last but not least: How reproducible are the distributions in the emulsions? And how reliably can the freezing spectra be evaluated (as e.g., those shown in Fig. 4)? And what is the uncertainty of the derived values?*

The homogeneity of lanolin-mineral oil mixture determines the reproducibility and stability of the size distribution of droplets in emulsions as discussed previously in *"Sect. A2 Uncertainty in the calculation of water droplet size distribution"* in Kaufmann et al. (2016). We have shown that the

fraction of ice active particles varies at most by a factor of 2 due to the changes in the water droplet size distribution for emulsions prepared with a batch of lanolin-mineral oil mixture and dust suspension with a few months gap. This result is in accordance with the comparison of the droplet size distributions measured at different times.

$F_{het}$ is evaluated from freezing experiments with at least two separate emulsions prepared from each suspension. Deviations are on average ~2 % from the mean. The maximum observed deviation from the mean was 10 %. This has been discussed in Sects. 2.2 (Lines 165-167) and 3.2 of the manuscript.

*line 345: You observe that $F_{het}$ decreases during the first day after microcline was suspended in pure water, while $T_{het}$ was preserved. Could a reason be that droplets are settling out? Again, it seems that the majority of your droplets might act different than the few ones that determine the freezing onset. And in this respect, if the different emulsions had different surface area distributions for the experiments in pure water and in dilute $NH_3$ and $(NH_4)_2SO_4$ solutions, this may also explain observed differences. Please discuss this shortly, too.*

We age the microcline suspensions and not the emulsions. Emulsions are freshly prepared from the suspensions right before running a freezing experiment. Settling of droplets in the emulsion overnight is therefore not an issue. We test the stability of the emulsion during the measurement in the DSC by performing a run with a cooling rate of 10 K/min before and after the run with a cooling rate of 1 K/min to check whether the DSC curve is the same for the first and the third cooling cycle. We use the second cooling cycle for evaluation. This is described in Sect. 2.1.

We have shown in Appendix B of the manuscript that particle aggregation does not explain our observations. There, we determined the particle size distribution of microcline suspended in pure water/solutions with a laser diffraction particle size analyzer. The only case where microcline shows slight aggregation is for the suspensions in 0.5 wt% KCl solution. Re-aggregation within droplets after emulsion preparation is possible in this case, but it is not sufficient to explain the drastic loss of IN activity shown in Figs. 3 and 5.

*As a bottom line of all of that said above (and besides for revisions in the text mentioned above), the direct translation of your results to the atmosphere, even with giving degrees of Kelvin by which the occurrence of freezing may be shifted, needs to be discussed more critically or may even be shortened. Your results on the influence on the surfaces and surface sites alone is already a valuable contribution.*

The intention of the "Atmospheric Implication" section is to show that the chemical-exposure history is relevant for the IN activity of microcline. We identify here a case for which a condensation freezing scenario results in a higher freezing temperature than an immersion freezing scenario. The focus is on the difference in freezing temperatures between condensation and immersion freezing. The exact values are not important. To make this clear, we add to the revised manuscript starting on line 473 of the revised manuscript: "As the basis of this discussion we use the freezing onsets and heterogeneously frozen fractions shown in Figs. 3 and 5, respectively. Figure 9 therefore shows freezing onsets that reflect the IN activity of the best microcline particle out of about 1000. Scenarios for average microcline particles would be exactly the same, only with all reported temperatures shifted downwards to about the maximum of the heterogeneous freezing signal, i.e. by 2 – 4 K." (Lines 473-476)

*line 51ff: Below, you discuss that deposition ice nucleation was questioned by Marcolli (2014), and similarly, for condensation freezing, you should also include that Vali et al. (2015) says: "Whether condensation freezing on a microscopic scale, if it occurs, is truly different from deposition nucleation, or distinct from immersion freezing, is not fully established."*

We mention this statement in the revised manuscript starting on line 54: "Although condensation freezing is usually mentioned as a distinct mode, it is still debated whether, on a microscopic scale, it is really distinct from the other freezing modes (Vali et al., 2015)."

*This is also related to line 442, where you use "condensation freezing", which, however, following the definition by Vali et al. (2015) given in your introduction does certainly not take place in your DSC measurements. But you used these measurements to make up your scenarios. Please be consistent!*

Indeed, our definition of condensation freezing is inaccurate. Thank you for pointing this out. We improve it in the revised manuscript by reformulating the definition of condensation freezing on line 53: "Condensation freezing is considered to take place when IN is concurrent with cloud droplet activation." With this improved definition, the scenario fits the definition of condensation freezing.

*line 276-278: Is it really so improbable that the (100) surface is exposed? - After all, you milled your samples, and the number density of active sites in microcline is "only" ~1000000 cm-3 at 251 K in Atkinson et al. (2013). This site density would need to be compared to the expected number of cracks and defaults that may occur during milling, before this statement you make here can be made. BTW:*

*this is again a point where it would be good to know the exact distributions of droplet sizes and of particles in the droplets.*

Cleavage upon milling occurs mostly along low energy faces. According to Pedevilla et al. (2016), the most easily cleaved face (001) also shows IN activity. The high energy (100) surface is presumably most important at the highest observed freezing temperatures which are not probed in our experiments, while the most common (001) faces become more and more important at lower freezing temperatures. We improve our argumentation in the revised manuscript, line 323: "If this were the case, the (100) face would need to be present on almost all particles since the majority of submicron microcline particles show IN activity (Niedermeier et al., 2015; Kaufmann et al., 2016; Burkert-Kohn et al., 2017). However, the (100) face is a high energy face that is not easily cleaved during milling and disappears during crystal growth. Moreover, Pedevilla et al. (2016) attribute IN activity to the common, easily cleaved (001) face, which we expect to dominate IN on submicron particles due to its prevalence on the particle surface." (Lines 323-327)

*The whole chapter 4.5: This chapter made a somewhat unfinished impression on me. How do the results here fit in line with what you described earlier? And as you only did the experiments at a single (and always different) pH for each of the solutions, a difference between the effects of the dissolved substance versus the pH cannot be obtained. This should also be mentioned in the text (e.g., connected to what you write in line 381). Also, in the end of Chapter 4.5, you cite a number of studies, however, without putting them in context to your results, so while reading this part of your text, I got confused. Similarly, when later reading the part on aging in the conclusions (line 501 ff) I was astounded as this did not reflect what I took from this chapter. Please revise these parts of the text.*

The aim of the reversibility tests are twofold, namely (i) to learn more about the properties of active sites by testing their stability under severe pH conditions, (ii) to investigate the relevance of microcline as an INP after atmospheric aging. We present them as a function of pH, because we think that increased surface dissolution at low and high pH limits IN activity of samples aged under severe pH conditions. To make our reasoning clearer in the revised manuscript we add to the revised manuscript on line 419: "The aim of the reversibility tests are twofold, namely (i) to learn more about the properties of active sites by testing their stability under severe pH conditions, (ii) to investigate the relevance of microcline as an INP after atmospheric aging." (Lines 419-421)

The effect of aging (hence, surface dissolution) depends on pH since the dissolution of feldspars is slowest at neutral or near neutral conditions (pH 3 – 8) and increases towards low and high pH (Helgeson et al., 1984). Burkert-Kohn et al. (2017) showed strong reduction in IN efficiency of 300 nm K-feldspar particles after 12 h of exposure to 1 molar sulfuric acid and nitric acid. Therefore, in Fig. 8 we use the pH as an ordering parameter rather than stating that it is a strict function of pH. Starting on line 425, we add to the revised manuscript: "We chose a presentation as a function of pH because we consider the increased dissolution rates at low and high pH (see Sect. 4.5) as a determinant of IN activity after aging because it enhances the degradation of the surface which may result in the irreversible loss of active sites." (Lines 425-427)

We shortened the discussion of studies from the literature and moved it to Sect. 4.5 of the revised manuscript treating the aging of microcline.

Moreover, we went critically through the whole section and improved the formulation. We hope that with these modifications, it is now clear that lines 501-504 of the conclusions (1$^{st}$ submission ACPD version) give just a short summary of the findings from the aging and recovery experiments.

*line 410: You state that "Saharan dust particles undergo little chemical processing during long-range transport across the Atlantic unless they become incorporated in cloud droplets". However, dust particles are CCN in the atmosphere (Karydis et al., 2011), so I wonder if you want to say that dust particles do not act as CCN, or that they do not become incorporated in cloud droplets because Saharan air masses are so dry that clouds do not form? Please clarify, and make clear that dust particles are CCN.*

We agree that dust particles, especially coarse mode, act as CCN during transport across the Atlantic Ocean. We have improved our statement in the revised manuscript (Lines 457-459): "Furthermore, airborne and ground station measurements imply that Saharan dust particles undergo little chemical processing during long-range transport across the Atlantic unless they become incorporated in cloud droplets by acting as cloud condensation nuclei (CCN)." With this sentence, we summarize results from field studies by Matsuki et al. (2010), Denjean et al. (2015), and Fitzgerald et al. (2015).

*line 435: (Again:) Your method prohibits to make statements about single particles - and, strictly speaking, also about atmospheric onset temperatures, as a single (and then likely smaller) particle in the atmosphere will only activate ice at lower temperature. I know that I mentioned this before. But again, I urge you to state this clearly.*

Niedermeier et al. (2015) have shown that the freezing onsets lie in the range of 248 K – 250 K for individual 200 nm – 500 nm microcline particles. This is only 2 – 4 K lower than the freezing onset that we observe in our experiments. If we took the maximum of the heterogeneous freezing signal that would correspond better with the average/median freezing temperature of the particles ($T_{50}$), this would not change the discussion of the atmospheric scenarios. We would like to stick to the heterogeneous freezing onsets because this is the value that we report throughout the study. The scenarios discussed in Fig. 9 should be considered as general guidelines for the effect of solutes on microcline particles.

To make this clear, we add to the revised manuscript starting on line 473 of the revised manuscript: "As the basis of this discussion we use the freezing onsets and heterogeneously frozen fractions shown in Figs. 3 and 5, respectively. Figure 9 therefore shows freezing onsets that reflect the IN activity of the best microcline particle out of about 1000. Scenarios for average microcline particles would be exactly the same, only with all reported temperatures scaled down to about the maximum of the heterogeneous freezing signal, i.e. by 2 – 4 K." (Lines 473-476)

*line 505: Let me ask you a question: Have you ever observed ice crystals outside of clouds in the mixed phase cloud temperature range (unless they fall out from a cloud)? What you suggest here might suggest that this could be observed. Based on the fact that your method rather is a bulk method and not one for single particles, I (again) suggest you are careful in drawing conclusions for processes going on in the atmosphere.*

We do not suggest anything like this on line 505 (1st submission ACPD version). Maybe the reviewer's comment is related to "mixing nucleation" that we mention on line 475 (1st submission ACPD version). We are referring here to conditions below water saturation but above ice saturation. Under such conditions, ice nucleation can be triggered by drying an air parcel especially at the periphery of clouds. We agree that our conclusions still should be confirmed by single particle measurements in continuous flow diffusion chambers.

*line 45: The abbreviation "IN" is used but has only been defined in the abstract. I'd define it again on the first appearance in the text.*

The abbreviation "IN" has been now defined on the first appearance (line 42) in the revised manuscript.

*line 72: You say "the particle", but in this context, it is not clear, which particle you mean.*

The subject of this line are particles exceeding 1 µm. The suggested line has been modified to "….such supermicron particles with excellent active sites and hence a high IN efficiency." (Lines 75-76)

*line 115: Replace "the" with "a", as this is where the setup is first introduced. Also, DSC was defined in abstract, but I'd define it again in main text upon its first appearance, ideally together with a citation where it is described in detail.*

We agree with the reviewer's suggestion. We define DSC on line 96 and write on line 140 of the revised manuscript: "Immersion freezing experiments were carried out with a DSC (Q10 from TA instruments) (see Zobrist et al. (2008) for details)."

*line 115: Upon reading this the first time, I wished for more information on the microcline when it was first mentioned here, particularly as you give all the detailed information about all the chemicals here, too. Now I know that this is given in 2.4. – maybe you could swap the chapters, so that 2.4 comes first, or you could at least mention here that there is more on the microcline sample later.*

We agree with the reviewer's suggestion. Section 2.4 from the discussion paper has been moved up and changed to Section 2.1 in the revised manuscript. Sections 2.1, 2.2 and 2.3 from the discussion paper have been renumbered to Sects. 2.2, 2.3 and 2.4, respectively, of the revised manuscript.

*line 120: Again, this is the first time that emulsions are mentioned, so delete "the".*

Suggested change has been made.

*line 155: Delete the "," following "Microcline".*

Suggested change has been made.

*line 179-180: There is something wrong with this sentence, please correct. Looks like a copy/paste error to me.*

We have reformulated this sentence in the revised manuscript, starting on line 191: "For 20 wt% microcline suspensions, droplets with diameters of 1.1µm are on average filled with one microcline particle, while smaller droplets are on average empty. This number shifts to 6.3 µm for 0.2 wt% microcline suspensions."

*line 275: Kiselev et al. was published online 2016, and I have a version downloaded 2017 that says "cite as ..., 2016" - please check which year is correct.*

We agree to the reviewer's suggestion. The concerned reference has been cited as "Kiselev et al., (2016)" in the revised manuscript.

*line 298: I'd prefer "suggest" to "conclude". BTW: In line 296, you say that "excess solute strength hampers the IN efficiency". Do you have any idea why that would be? If you could add a sentence on that here, I'd appreciate it.*

We changed "suggest" to "conclude" in the revised manuscript.

The hampering of IN efficiency at higher solute strength (hence, lower water activity) has been discussed in detail in Section 4.4 of the revised manuscript. The concerned line has been modified to "excess solute strength hampers the IN efficiency (discussed in Sect. 4.4)" in the revised manuscript. (Lines 344-345)

*line 310: Please add at or above which water activity you are referring to, here, in this sentence, as data all agree quite well in the lower concentration range.*

To specify the water activity range for enhanced $T_{het}$ yet hampered $F_{het}$, the concerned line has been modified in the revised manuscript to "Similarly, increasing the concentration of $NH_4HSO_4$ resulted in the reduction of $F_{het}$ while $T_{het}$ still showed an increase at low concentration ($a_w = 1 - 0.85$)." (Lines 355-356)

References

[revised manuscript text omitted]

---

## Author Response (AR2)

We thank reviewers and co-editor for their constructive comments and suggestions. We present below point-wise changes made in the revised manuscript.

1. ***Main comment from reviewer:*** *"This only remaining issue for me is that I'd wish that the following would be explicitly stated in the chapter "atmospheric implications" (not necessarily with these words, but with this content): For mixed phase clouds (i.e., those clouds appearing at > ~ -38°C), formation of droplets is observed before ice formation is observed (e.g., Ansmann et al., 2009; Wiacek et al., 2010; de Boer et al., 2011 and others). This is important and has implications on the relevance of the pathways you describe and should at least be mentioned or even discussed."*

We have added the above-mentioned discussion in the revised manuscript on lines 454-456: "The temperature regime probed in our experiments is relevant to mixed phase clouds (273 K - 236 K) where liquid phase is observed before ice crystal formation (Ansmann et al., 2009; Wiacek et al., 2010; de Boer et al., 2011)."

2. **Other changes made:**

- As per discussed with the co-editor via email, the title of the manuscript has been revised to "Ice nucleation activity of silicates and aluminosilicates in pure water and aqueous solutions. Part I - The K-feldspar Microcline"

- Few lines have been added to the *Introduction* about the current paper being a part of a series (lines 116-118): "In this and the companion papers (Part II and III) we attempt to relate IN activities of pure mineral surfaces with the mineral surface properties by investigating the differences in IN activity of structurally similar minerals in pure water and aqueous solutions."

[revised manuscript text omitted]